# SARE: SEMANTIC-AWARE RECONSTRUCTION ERROR FOR GENERALIZABLE AI-GENERATED IMAGE DETECTION

## ABSTRACT

Recently, AI-generated image detection has gained increasing attention, as the rapid advancement of image generation technologies has raised serious concerns about their potential misuse. While existing detection methods have achieved promising results, their performance often degrades significantly when facing fake images from unseen, out-of-distribution (OOD) generative models, since they primarily rely on model-specific artifacts and thus overfit to the models used for training. To address this limitation, we propose a novel representation, namely Semantic-Aware Reconstruction Error (SARE), that measures the semantic difference between an image and its caption-guided reconstruction. The key hypothesis behind SARE is that real images, whose captions often fail to fully capture their complex visual content, may undergo noticeable semantic shifts during the caption-guided reconstruction process. In contrast, fake images, which closely align with their captions, show minimal semantic changes. By quantifying these semantic shifts, SARE provides a robust and discriminative feature for detecting fake images across diverse generative models. Additionally, we introduce a fusion module that integrates SARE into the backbone detector via a cross-attention mechanism. Image features attend to semantic representations extracted from SARE, enabling the model to adaptively leverage semantic information. Experimental results demonstrate that the proposed method achieves strong generalization, outperforming existing baselines on benchmarks including GenImage and ForenSynths. We further validate the effectiveness of caption guidance through a detailed analysis of semantic shifts, confirming its ability to enhance detection robustness.

## 1 INTRODUCTION

In recent years, image generation technologies, such as Generative Adversarial Networks (GANs) (Goodfellow et al., 2014; Zhu et al., 2017; Brock et al., 2019; Karras et al., 2018) and Diffusion Models (DMs) (Ho et al., 2020; Song et al., 2021; Rombach et al., 2022; Nichol et al., 2022), have made remarkable progress, enabling the synthesis of highly realistic images that are often indistinguishable from real images. This realism has raised growing concerns about potential misuse, particularly in generating harmful or deceptive content (Ferreira et al., 2020; Juefei-Xu et al., 2022). To address these risks, developing reliable methods for detecting AI-generated images has become increasingly important.

A common approach in existing detection methods is to train a binary classifier using real and fake images sourced from a finite set of generative models available during training (Bayar & Stamm, 2016; Wang et al., 2020; Liu et al., 2020; Wang et al., 2023). While these detectors typically exhibit strong performance when test images are generated by the same models used during training, their performance often drops significantly in real-world scenarios, where they inevitably encounter fake images from unseen generative models that are not included in the training data (Zhang et al., 2019; Luo et al., 2021; Yan et al., 2023). To ensure robustness in practical deployment, it is essential to develop detection methods that can generalize effectively to such unseen and out-of-distribution (OOD) generative models.

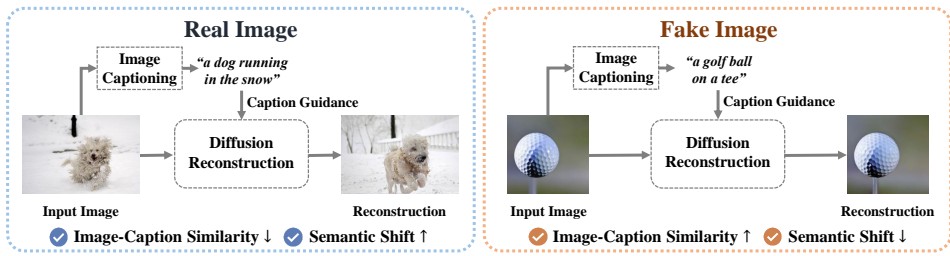

Figure 1: Comparison of caption-guided reconstructions for real and fake images. Real images, whose captions often fail to fully capture their complex visual content, undergo noticeable semantic shifts during caption-guided reconstruction. In contrast, fake images, which align closely with their captions, tend to exhibit minimal semantic changes.

Recent studies have proposed several strategies to address the generalization challenges inherent in generated image detection. These strategies include training methods such as reconstruction-based learning (Wang et al., 2023; Luo et al., 2024; Chu et al., 2025) and data augmentation (Chen et al., 2024), as well as architectural approaches (Ojha et al., 2023; Wu et al., 2023; Tan et al., 2025) that leverage a large pre-trained model like CLIP (Radford et al., 2021). Despite these advances, the robustness of existing methods remains limited, as they primarily focus on identifying visual artifacts introduced during the generative process (Frank et al., 2020; Wang et al., 2020; 2023; Chen et al., 2024). Due to the distinct characteristics of different generative models, such artifacts are inherently model-specific and fail to generalize across diverse models (Luo et al., 2021; Corvi et al., 2023; Ojha et al., 2023). As a result, approaches that rely on these artifacts tend to overfit to the models used for training, which leads to degraded performance in OOD scenarios.

To overcome these limitations, we explore a fundamental property commonly observed in fake images. Prior work (Sha et al., 2023) has shown that the similarity between fake images and captions generated by an image-captioning model is typically higher than that of real images. Real images contain complex, fine-grained details that short captions cannot cover, whereas fake images include only the elements explicitly specified in the user's text prompt. Inspired by this observation, we hypothesize that the relationship between an image and its caption reflects a general characteristic of fake images, providing a robust signal for detection across diverse generative models.

In this paper, we propose Semantic-Aware Reconstruction Error (SARE), a novel representation for detecting AI-generated images that measures the semantic difference between an image and its reconstruction. Specifically, we introduce a caption-guided reconstruction pipeline to effectively leverage the relationship between an image and its caption in the detection process. The key idea is that real images, which often exhibit low similarity to their captions, may undergo noticeable semantic shifts during caption-guided reconstruction. In contrast, fake images, whose content is well captured by their captions, show minimal semantic shifts. As shown in Figure 1, the real image is reconstructed into a noticeably different dog since the caption provides only a coarse description (e.g., "a dog running in the snow") without capturing fine details such as the dog's breed, pose, or background. Conversely, the fake image of a golf ball remains largely unchanged after reconstruction, as its content can be sufficiently described by a simple caption. By capturing these fundamental differences between real and fake images, SARE provides a discriminative and generalizable feature for detecting fake images across diverse generative models. Additionally, we design a fusion module that integrates SARE into the backbone detector via a cross-attention mechanism. The original image features attend to the semantic representations extracted from SARE, allowing the model to adaptively incorporate semantic information.

We validate the effectiveness of SARE through extensive experiments on the GenImage (Zhu et al., 2023) and ForenSynths (Wang et al., 2020) datasets. The proposed framework significantly improves the performance of the backbone model across both seen and unseen generators, achieving the best average results compared to existing detection methods. The results demonstrate the robustness of SARE in OOD scenarios, confirming its strong generalization to fake images from diverse generative models.

## 2 RELATED WORK

### 2.1 DETECTION BASED ON IMAGE CAPTION

Caption-based detection methods explore the use of image captions as a cue for detecting generated images. DE-FAKE (Sha et al., 2023) finds that generated images tend to align more closely with their captions compared to real images. Based on the observation, it adopts separate encoders for image and caption to exploit the relationship between them. Following this direction, C2P-CLIP (Tan et al., 2025) proposes a method that injects category-level prompts to enhance detection performance. LASTED (Wu et al., 2023) introduces a language-guided contrastive learning framework that leverages textual labels to improve generalization.

### 2.2 DETECTION BASED ON IMAGE RECONSTRUCTION

Reconstruction-based detection methods utilize a pre-trained diffusion model to reconstruct the input image and analyze the differences between the original and reconstructed images. DIRE (Wang et al., 2023) introduces reconstruction error as the discriminative feature for detection, based on the assumption that fake images can be reconstructed more accurately than real images. To improve efficiency, LaRE (Luo et al., 2024) computes this reconstruction error in the latent space using a single-step denoising process, substantially reducing computational cost while preserving detection performance. DRCT (Chen et al., 2024), rather than relying on reconstruction error, treats reconstructed images as hard samples and adopts a contrastive learning framework to facilitate discriminative feature learning. FakeInversion (Cazenavette et al., 2024) not only exploits the reconstructed images but also incorporates additional feature maps derived from caption-conditioned DDIM inversion (Song et al., 2021), where captions are mainly employed to stabilize the inversion and reconstruction process. In contrast, our method SARE explicitly leverages the relationship between an image and its caption. Motivated by the observation that fake images tend to exhibit higher similarity to their captions than real images, SARE quantifies the semantic difference between the image and its caption-guided reconstruction. This semantic-aware discrepancy serves as a robust detection signal, enabling SARE to generalize effectively across diverse generative models.

## 3 PROPOSED METHOD

### 3.1 MOTIVATION

Existing methods (Frank et al., 2020; Wang et al., 2020; 2023; Chen et al., 2024) for detecting fake images primarily rely on visual artifacts or traces left by the generative models. A representative example is DIRE (Wang et al., 2023), which reconstructs the input image with a pre-trained diffusion model and leverages the pixel-wise reconstruction error as a discriminative feature for classification. It is based on the assumption that fake images exhibit smaller reconstruction errors than real images, as both the original and reconstructed images belong to the same generative distribution and thus share similar visual patterns. However, our empirical observation suggests that this assumption often does not hold in OOD scenarios, where fake images are synthesized by unseen generators that were not available during training. As shown in Figure 2, when Stable Diffusion v1.4 (Rombach et al., 2022) is used for reconstruction, fake images from unseen models such as ADM (Dhariwal & Nichol, 2021) or Big-GAN (Brock et al., 2019) produce much larger reconstruction errors, even exceeding those of real images. This implies that diverse generative models,

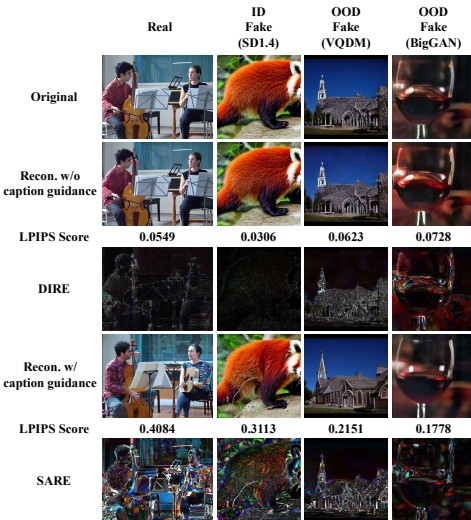

Figure 2: Examples from the GenImage dataset (Zhu et al., 2023) and their corresponding DIREs (Wang et al., 2023), SAREs, and LPIPS scores. Images are reconstructed using Stable Diffusion v1.4, and the pixel values of the DIREs and SAREs are scaled by 2 for clearer visualization.

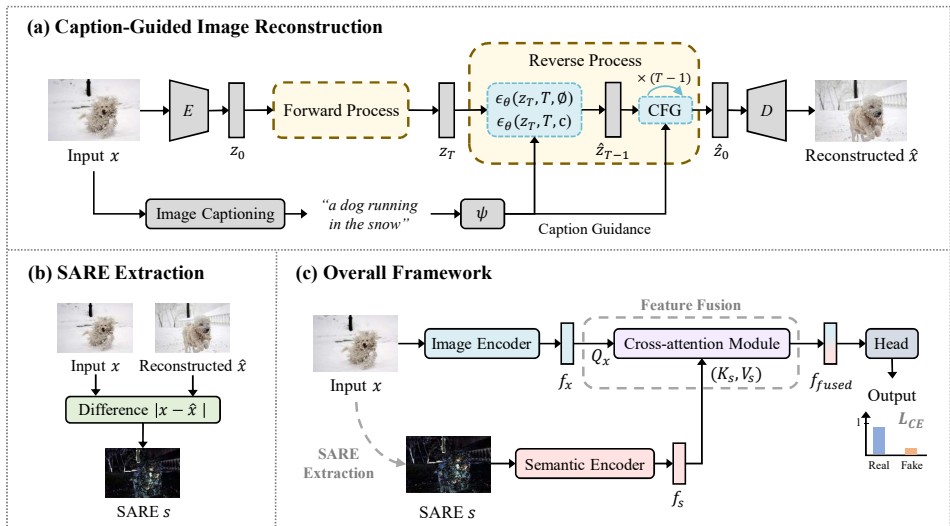

Figure 3: Overview of the SARE framework. Our method reconstructs the input image conditioned on its caption using the Stable Diffusion model with classifier-free guidance. SARE is computed as the difference between the input and reconstructed image, and is incorporated into the detection process through a cross-attention module that leverages image features as queries and SARE features as keys and values. The pixel values of the SARE are scaled by 2 for clearer visualization.

including the reconstruction model and unseen generators, exhibit distinctive characteristics and traces. From this observation, we suggest that methods relying on visual artifacts from a specific generation process may struggle to generalize in OOD scenarios. This limitation highlights the need for more generalizable detection cues that can perform reliably across diverse generative models.

## 3.2 SEMANTIC-AWARE RECONSTRUCTION ERROR

We propose Semantic-Aware Reconstruction Error (SARE), a novel detection feature designed to enhance generalization in AI-generated image detection. The hypothesis of SARE is that the relationship between an image and its caption may reflect fundamental differences between real and fake images, and thus serve as a generalizable detection cue. SARE aims to effectively leverage this property by introducing a caption-guided reconstruction framework. The framework consists of three main steps: (1) image captioning, (2) caption-guided image reconstruction, and (3) SARE extraction.

**Image Captioning**   For a given image $x$, we utilize a pre-trained image captioning model to generate a descriptive caption $C$. This caption $C$ is used as the text condition for the subsequent reconstruction process.

**Caption-guided Image Reconstruction**   Given the caption $C$, we reconstruct the input image $x$ by using a pre-trained text-conditional diffusion model. Specifically, we leverage the Stable Diffusion model (Rombach et al., 2022) with classifier-free guidance (Ho & Salimans, 2021). The input image $x$ is first encoded into a latent representation $z_0$ using the Variational Autoencoder (VAE) encoder (Kingma & Welling, 2014). The forward process then adds Gaussian noise to $z_0$ following a predefined noise schedule. The noisy latent at a given timestep $t$ is computed as:

$$z_t = \sqrt{\bar{\alpha}_t}z_0 + \sqrt{1 - \bar{\alpha}_t}\epsilon, \tag{1}$$

where $\epsilon \sim \mathcal{N}(0, \mathbf{I})$, and $\bar{\alpha}_t = \prod_{s=1}^{t} \alpha_s$. The *strength* parameter determines the amount of noise added during reconstruction. The number of forward diffusion steps is set to $T = \lfloor strength \times T_{\max} \rfloor$, where $T_{\max}$ is the total number of diffusion steps.

Starting from the noisy latent $z_T$, the reverse process aims to obtain $\hat{z}_0$ through an iterative denoising process conditioned on the caption $C$. At each denoising step, the noise prediction network

$\epsilon_\theta(z_t, t, c)$ estimates the noise $\epsilon$, where $c = \psi(C)$ denotes the caption embedding obtained from the CLIP text encoder (Radford et al., 2021). We adopt classifier-free guidance, which combines the conditional and unconditional noise predictions as follows:

$$\epsilon_\theta(z_t, t, c, \varnothing) = w\epsilon_\theta(z_t, t, c) + (1 - w)\epsilon_\theta(z_t, t, \varnothing), \tag{2}$$

where $w$ is the guidance scale and $\varnothing = \psi(\text{""})$ denotes the null text embedding. The denoising process using DDIM sampling (Song et al., 2021) can be represented by:

$$z_{t-1} = \sqrt{\alpha_{t-1}} \frac{z_t - \sqrt{1 - \alpha_t}\,\epsilon_\theta(z_t, t, c, \varnothing)}{\sqrt{\alpha_t}} + \sqrt{1 - \alpha_{t-1}}\epsilon_t, \tag{3}$$

where $\alpha_{t-1} = \frac{\bar{\alpha}_{t-1}}{\bar{\alpha}_t}$ and $\epsilon \sim \mathcal{N}(0, \mathbf{I})$, for $t = T, ..., 1$. After $T$ denoising steps, the final latent $\hat{z}_0$ is obtained and decoded by the VAE decoder to produce the reconstructed image $\hat{x}$.

**SARE Extraction**    Once we obtain the original image $x$ and the reconstructed image $\hat{x}$, we compute the SARE by measuring the difference between the two images. SARE is defined as follows:

$$\text{SARE}(x, \hat{x}) = |x - \hat{x}|, \tag{4}$$

where $|\cdot|$ denotes the absolute value. SARE quantifies the semantic changes introduced during the caption-guided reconstruction process. Since real images often contain complex visual details that cannot be fully reflected in their captions, their reconstructions result in noticeable semantic shifts. In contrast, fake images typically align closely with their captions and therefore tend to undergo minimal semantic changes. By capturing these differences between real and fake images, SARE can serve as a discriminative feature for robust detection across diverse generative models.

### 3.3 FUSION MODULE

We propose a fusion module to effectively integrate SARE into the detection process. Given an input image $x$ and its corresponding SARE $s$, we extract the image feature $f_x$ and the semantic feature $f_s$ using the image encoder $E_x$ and the semantic encoder $E_s$, respectively:

$$f_x = E_x(x), \; f_s = E_s(s). \tag{5}$$

To obtain the fused feature $f_{fused}$, we employ a cross-attention mechanism by leveraging $f_x$ for query and $f_s$ for key and value as follows:

$$Q_x = f_x W_Q, \; K_s = f_s W_K, \; V_s = f_s W_V,$$
$$f_{fused} = CrossAttn(Q_x, K_s, V_s), \tag{6}$$

where $W_Q, W_K$, and $W_V$ are the linear projections for the query, key, and value, respectively. This fused representation allows the model to incorporate semantic information as an additional cue. Subsequently, $f_{fused}$ is passed through a fully connected layer that serves as the classification head, and the model is trained using binary cross-entropy loss.

## 4 EXPERIMENT

### 4.1 EXPERIMENTAL SETTINGS

**Datasets and Evaluation Metrics**    We evaluated the performance of detection models using the GenImage (Zhu et al., 2023) dataset, which is divided into 8 subsets. Each subset consists of real images from ImageNet Deng et al. (2009) and fake images synthesized by a single generative model. The generative models are Midjourney (MJ) (Mid, 2022), Stable Diffusion v1.4&v1.5 (SDv1.4&v1.5) (Rombach et al., 2022), ADM (Dhariwal & Nichol, 2021), GLIDE (Nichol et al., 2022), Wukong (Wuk, 2022), VQDM (Gu et al., 2022), and BigGAN (Brock et al., 2019). We used the training split from the SDv1.4 subset for training, and the test splits from all subsets for evaluation. For cross-dataset evaluation, we trained the models on the SDv1.4 subset of GenImage and evaluated them on the ForenSynths (Wang et al., 2020) test set. The ForenSynths test set contains 11 subsets, where each subset comprises real images from the training data of a specific generative model and fake images produced by that model. The generative models in ForenSynths include

| Method | MJ | SDv1.4 | SDv1.5 | ADM | GLIDE | Wukong | VQDM | BigGAN | Avg ACC.(%) |
|---|---|---|---|---|---|---|---|---|---|
| GramNet | 73.32 | 96.73 | 96.55 | 51.73 | 58.85 | 91.19 | 57.05 | 48.63 | 71.76 |
| Conv-B | 84.59 | **100.00** | **99.91** | 52.86 | 57.14 | **99.88** | 58.77 | 50.01 | 75.40 |
| UnivFD | 89.56 | 96.94 | 96.56 | 57.20 | 71.12 | 95.03 | 68.67 | 57.83 | 79.11 |
| DIRE | 51.03 | 99.96 | **99.91** | 51.78 | 59.26 | 99.79 | 50.18 | 50.88 | 70.35 |
| DE-FAKE | 85.55 | 97.93 | 97.82 | 53.53 | 65.28 | 91.57 | 55.98 | 49.16 | 74.60 |
| DRCT | **90.89** | 94.75 | 94.28 | 78.54 | 87.52 | 94.58 | 90.12 | 79.76 | 88.81 |
| SARE (ours) | 90.32 | 97.21 | 97.04 | **84.47** | **93.55** | 97.05 | **93.66** | **92.05** | **93.17** |

Table 1: Accuracy (ACC, %) comparisons of different detectors on the GenImage dataset (Zhu et al., 2023). All methods are trained on the SDv1.4 subset and evaluated across 8 subsets. The best and second-best results are indicated in **bold** and underlined, respectively.

| Method | MJ | SDv1.4 | SDv1.5 | ADM | GLIDE | Wukong | VQDM | BigGAN | Avg AUC.(%) |
|---|---|---|---|---|---|---|---|---|---|
| GramNet | 91.54 | 99.56 | 99.49 | 69.87 | 83.52 | 98.10 | 78.40 | 39.36 | 82.48 |
| Conv-B | **99.54** | **100.00** | **99.94** | 90.10 | 96.72 | **100.00** | 93.82 | 86.61 | 95.84 |
| UnivFD | 97.54 | 99.57 | 99.51 | 73.09 | 89.46 | 98.99 | 87.53 | 79.19 | 90.61 |
| DIRE | 78.65 | **100.00** | **99.94** | 71.45 | 90.42 | 99.99 | 62.49 | 61.12 | 83.01 |
| DE-FAKE | 97.13 | 99.81 | 99.80 | 70.95 | 89.26 | 98.52 | 78.48 | 57.60 | 86.44 |
| DRCT | 96.91 | 99.64 | 99.52 | 88.47 | 94.61 | 99.42 | 96.44 | 90.30 | 95.66 |
| SARE (ours) | 96.83 | 99.94 | 99.93 | **94.87** | **98.00** | 99.83 | **98.31** | **97.51** | **98.15** |

Table 2: AUC (%) comparisons of different detectors on the GenImage dataset (Zhu et al., 2023). All methods are trained on the SDv1.4 subset and evaluated across 8 subsets. The best and second-best results are indicated in **bold** and underlined, respectively.

ProGAN (Karras et al., 2018), StyleGAN (Karras et al., 2019), BigGAN (Brock et al., 2019), CycleGAN (Zhu et al., 2017), StarGAN (Choi et al., 2018), GauGAN (Park et al., 2019), CRN (Chen & Koltun, 2017), IMLE (Li et al., 2019), SITD (Chen et al., 2018), SAN (Dai et al., 2019), and Deepfake (Rossler et al., 2019). For evaluation metrics, we employed accuracy (ACC) and the Area Under the ROC curve (AUC). Accuracy was computed with a fixed threshold of 0.5, following the baseline settings Wang et al. (2023); Chen et al. (2024).

**Implementation Details** To obtain reconstructed images for SARE and for the baseline models DIRE (Wang et al., 2023) and DRCT (Chen et al., 2024), we used SDv1 as the reconstruction model. For SARE, captions were generated using a pre-trained BLIP model (Li et al., 2022). Each caption was used to guide the reconstruction process, where we set the strength parameter to 0.5, the guidance scale to 7.5, and the maximum number of diffusion steps to 50. We adopted DRCT as the backbone detector, which utilizes CLIP:ViT-L/14 (Radford et al., 2021) as the image encoder. For the semantic encoder, we employed a ResNet50 model (He et al., 2016). During training, we applied random cropping and several augmentations, including horizontal flipping, Gaussian noise injection, Gaussian blurring, random rotation, JPEG compression with random quality, random scaling, grid dropout, and brightness and contrast adjustments. At test time, images were center-cropped. All models were designed to take input images of size $224 \times 224$. For SARE extraction, images were resized to 512 on the longer side before reconstruction, and the resulting SARE representations were fed into the encoder at a size of $224 \times 224$. We trained our proposed model for 17 epochs with a batch size of 512 and used the AdamW optimizer (Loshchilov & Hutter, 2019) with an initial learning rate of $1 \times 10^{-4}$.

### 4.2 COMPARISONS TO EXISTING DETECTORS

Tables 1 and 2 report the accuracies and AUC scores of different detection methods on the GenImage dataset. We compared our method with several detectors, including GramNet Liu et al. (2020), Conv-B (Liu et al., 2022), UnivFD (Ojha et al., 2023), DIRE, DE-FAKE (Sha et al., 2023), and DRCT. All models were trained on the SDv1.4 subset. For DE-FAKE, we used BLIP for caption-

| Method | Pro-GAN | Style-GAN | Big-GAN | Cycle-GAN | Star-GAN | Gau-GAN | CRN | IMLE | SITD | SAN | Deep-Fake | Avg ACC.(%) |
|---|---|---|---|---|---|---|---|---|---|---|---|---|
| GramNet | 49.20 | 48.57 | 49.73 | 48.91 | 49.05 | 48.70 | 47.59 | 47.50 | 65.56 | 57.99 | 58.02 | 51.89 |
| Conv-B | 54.66 | 50.47 | 52.50 | 50.03 | 49.47 | 50.19 | **49.94** | 52.50 | 62.5 | 66.44 | **80.19** | 56.26 |
| UnivFD | 67.97 | 53.92 | 68.47 | 67.73 | **79.94** | 56.21 | 38.04 | 54.64 | 63.89 | 65.53 | 60.56 | 61.54 |
| DIRE | 50.06 | 50.03 | 49.88 | 49.94 | 50.05 | 49.97 | 49.44 | 49.59 | 53.89 | 73.29 | 52.58 | 52.61 |
| DE-FAKE | 51.20 | 48.39 | 52.88 | 51.49 | 63.81 | 49.02 | 49.46 | 47.31 | 53.89 | 65.30 | 51.77 | 53.14 |
| DRCT | 74.59 | 67.41 | 83.10 | **92.40** | 62.23 | 78.89 | 41.67 | 51.86 | 66.11 | 79.45 | 55.78 | 68.50 |
| SARE (ours) | **84.44** | **76.32** | **83.17** | 90.24 | 59.58 | **81.28** | 46.6 | **60.94** | 61.39 | **85.16** | 51.54 | **70.97** |

Table 3: Accuracy (ACC, %) comparisons of different detectors under cross-dataset evaluation. All detectors are trained on the SDv1.4 subset of the GenImage dataset (Zhu et al., 2023) and evaluated on the ForenSynths test set (Wang et al., 2020). The best and second-best results are indicated in **bold** and underlined, respectively.

| Method | Pro-GAN | Style-GAN | Big-GAN | Cycle-GAN | Star-GAN | Gau-GAN | CRN | IMLE | SITD | SAN | Deep-Fake | Avg AUC.(%) |
|---|---|---|---|---|---|---|---|---|---|---|---|---|
| GramNet | 49.08 | 45.59 | 50.76 | 55.73 | 48.46 | 34.39 | 49.90 | 39.23 | 75.14 | 70.14 | 63.88 | 52.94 |
| Conv-B | 75.66 | 74.59 | 77.46 | 53.58 | 38.18 | 62.23 | 44.21 | **85.55** | 86.54 | **98.62** | **87.58** | 71.29 |
| UnivFD | 81.38 | 64.79 | 84.46 | 93.63 | 89.31 | 80.03 | 29.51 | 57.22 | 74.75 | 75.07 | 67.96 | 72.56 |
| DIRE | 55.64 | 52.37 | 45.25 | 47.64 | 51.94 | 45.38 | 43.86 | 62.73 | **93.95** | 98.44 | 84.34 | 61.96 |
| DE-FAKE | 55.74 | 46.53 | 70.09 | 76.11 | 71.15 | 43.10 | **51.76** | 46.21 | 51.93 | 77.38 | 51.11 | 58.28 |
| DRCT | 89.35 | 75.73 | **92.74** | **98.28** | **95.93** | 88.23 | 29.35 | 68.55 | 79.46 | 88.76 | 80.01 | 79.67 |
| SARE (ours) | **93.45** | **87.05** | 92.10 | 95.83 | 94.80 | **90.43** | 47.73 | 79.10 | 77.70 | 92.52 | 77.68 | **84.40** |

Table 4: AUC (%) comparisons of different detectors under cross-dataset evaluation. All detectors are trained on the SDv1.4 subset of the GenImage dataset (Zhu et al., 2023) and evaluated on the ForenSynths test set (Wang et al., 2020). The best and second-best results are indicated in **bold** and underlined, respectively.

ing, following the configuration described in the original paper. The results show that compared to DRCT, our method improves the average accuracy by 4.36%, and the average AUC by 2.49%, which indicates that integrating SARE effectively enhances the detection performance. Notably, our method achieves the highest average accuracy of 93.17% and AUC of 98.15%, outperforming all other detection approaches. While all the detectors show strong performance on SDv1.4, SDv1.5, and Wukong subsets, their performance tends to degrade significantly on other subsets like ADM, GLIDE, VQDM, and the non-diffusion model BigGAN. Our method maintains consistently high performance across all subsets, demonstrating robust generalization to diverse OOD generative models. Moreover, the proposed method outperforms DE-FAKE, suggesting that SARE leverages the relationship between an image and its caption more effectively than directly comparing image and caption embeddings obtained from CLIP.

## 4.3 CROSS-DATASET EVALUATION

To further assess the generalization ability of the detection methods, we conducted a cross-dataset evaluation. All detectors were trained on the SDv1.4 subset of the GenImage dataset and evaluated on the ForenSynths test set. Table 3 and Table 4 report the accuracy and AUC score of each method on this test set. Our method shows strong performance across diverse generative models, yielding an average accuracy of 70.97% and an average AUC of 84.40%, which are the highest among all detectors. These results highlight the effectiveness of our method in OOD scenarios, demonstrating its robust generalization to fake images from unseen generative models.

## 4.4 SEMANTIC SHIFT ANALYSIS

**Quantitative Results** To validate the core assumption that real images undergo larger semantic shifts than fake images, we measured the perceptual distance between an image $x$ and its reconstruction $\hat{x}$ using the Learned Perceptual Image Patch Similarity (LPIPS) (Zhang et al., 2018) metric. Figure 4a summarizes the average LPIPS scores for real and fake images in each subset of the GenImage dataset under two conditions: (1) reconstruction without caption guidance, and (2)

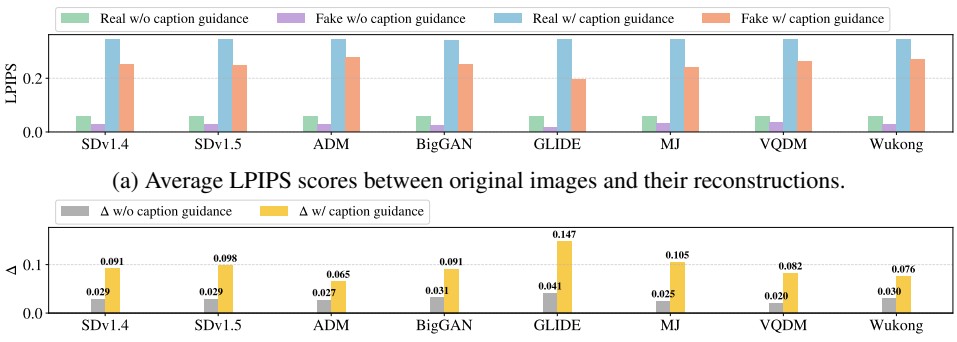

(a) Average LPIPS scores between original images and their reconstructions.

(b) $\Delta$ values measuring the LPIPS score gap between real and fake images.

Figure 4: Semantic shift analysis based on LPIPS scores (Zhang et al., 2018). Higher scores indicate lower similarity between the original and reconstructed images. Images are reconstructed under two conditions: with and without caption guidance.

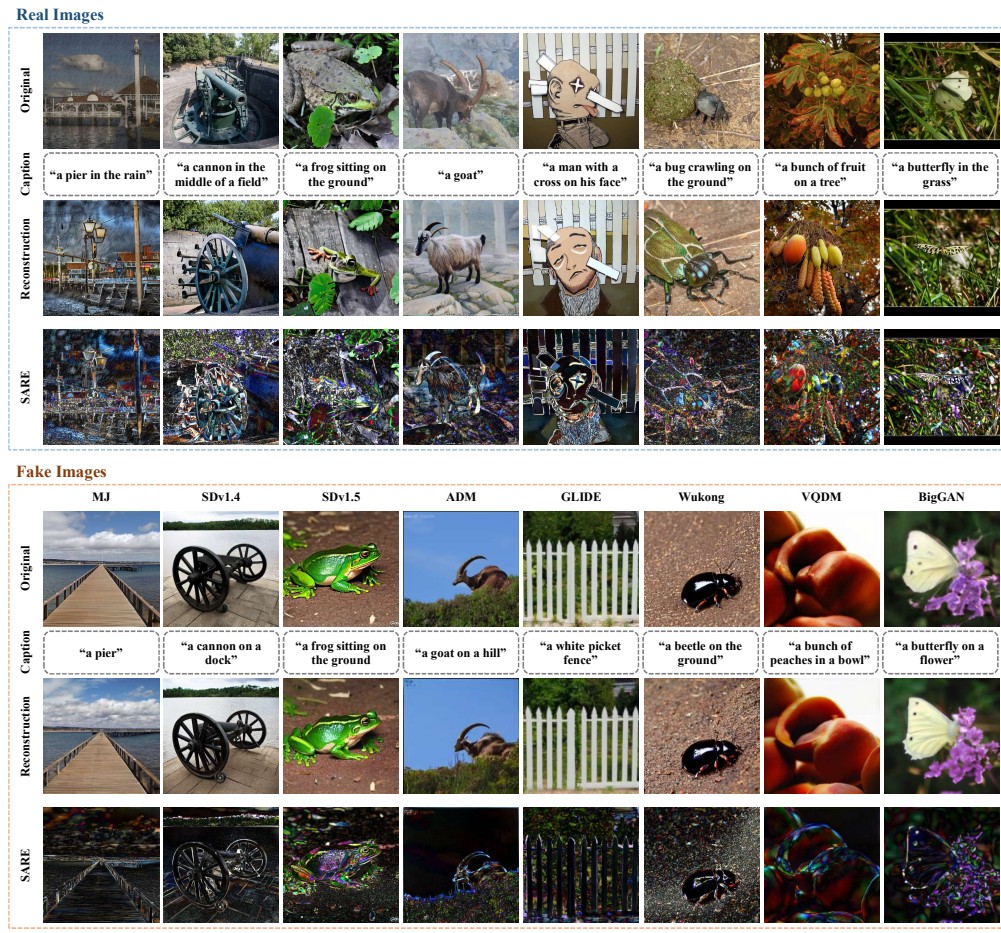

Figure 5: Real and fake images from the GenImage dataset (Zhu et al., 2023) with their captions generated by a pre-trained BLIP (Li et al., 2022), the corresponding reconstructions, and SAREs.

reconstruction with caption guidance. While real images consistently exhibit higher LPIPS scores than fake images in both settings, the gap between real and fake images is substantially larger when caption guidance is applied. To quantify this gap, we define $\Delta$ as follows:

$$\Delta = \mathbb{E}_{x \sim \mathcal{D}_{\text{real}}}[LPIPS(x, \hat{x})] - \mathbb{E}_{x \sim \mathcal{D}_{\text{fake}}}[LPIPS(x, \hat{x})]. \tag{7}$$

| Method | Image Captioning | Avg ACC.(%) | Avg AUC.(%) |
|---|---|---|---|
| DRCT | - | 88.81 | 95.66 |
| SARE (ours) | BLIP | **93.17** | **98.15** |
| | LLaVA-NeXT | 92.51 | 97.95 |

Table 5: Ablation study on the influence of image captioning models on the Gen-Image dataset (Zhu et al., 2023).

| Method | $w$ | Avg ACC.(%) | Avg AUC.(%) |
|---|---|---|---|
| DRCT | - | 88.81 | 95.66 |
| SARE (ours) | 2.5 | 93.15 | **98.24** |
| | 7.5 | **93.17** | 98.15 |
| | 12.5 | 93.04 | 98.13 |

Table 6: Ablation study on the guidance scale $w$ conducted on the GenImage dataset (Zhu et al., 2023).

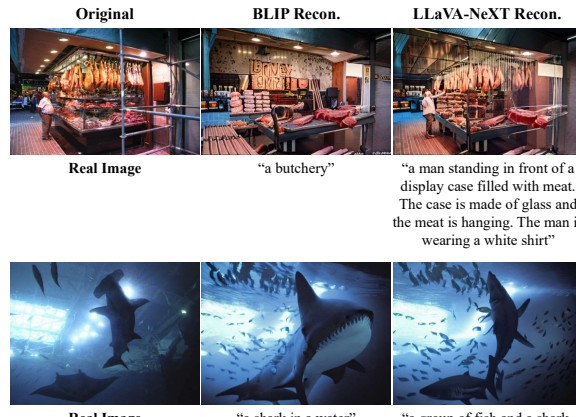

Figure 6: Real images from the GenImage dataset (Zhu et al., 2023) with captions from BLIP (Li et al., 2022) and LLaVA-NeXT (Liu et al., 2024), and their reconstructions.

As shown in Figure 4b, $\Delta$ is relatively small without caption guidance, but increases significantly in all subsets when caption guidance is used. These results suggest that the semantic difference between an image and its caption-guided reconstruction may serve as a more discriminative feature for detection, thereby leading to improved performance across diverse generative models.

**Qualitative Results and Visualizations** Figure 5 presents qualitative examples of real and fake images from the GenImage dataset and their caption-guided reconstructions, where captions were generated using a pre-trained BLIP. In GenImage, real images are sourced from ImageNet, while fake images are synthesized by generative models using ImageNet class labels as text prompts. For a fair comparison, we visualize real and fake images from the same ImageNet class label along with their reconstructions. The results show that real images tend to undergo larger semantic shifts than fake images during the caption-guided reconstruction process.

## 4.5 ABLATION STUDY

**Influence of Image Captioning models** To evaluate the impact of different image captioning models on detection performance, we conducted an ablation study using captions generated by pre-trained BLIP and LLaVA-NeXT-8B (Liu et al., 2024) on the GenImage dataset. As shown in Table 5, SARE demonstrates strong performance with both captioning models, but BLIP consistently achieves higher accuracy and AUC. To further examine this difference, Figure 6 visualizes real images from GenImage and their reconstructions guided by captions from the two models. BLIP tends to generate concise captions such as "a butchery", which do not fully capture fine details like "a man in a white shirt" or "hanging meat". As a result, the reconstructed images differ significantly from the original, leading to noticeable semantic shifts. In contrast, LLaVA-NeXT provides more detailed descriptions that include such elements, yielding reconstructions that remain relatively close to the input image and thus exhibit smaller semantic changes. These observations suggest that BLIP's coarse captions induce larger semantic shifts in real images, thereby enhancing the effectiveness of SARE in distinguishing real from fake images. A more detailed analysis of captioning models is presented in Appendix A.

**Influence of Guidance Scale** We investigated the impact of the guidance scale $w$ on detection performance within the caption-guided reconstruction framework. Table 6 presents the accuracy and AUC results on the GenImage dataset for different guidance scale values. The results show that incorporating SARE consistently improves the performance over the baseline across all settings. Notably, the best accuracy is achieved at $w = 7.5$ (93.17%), whereas the highest AUC is observed at $w = 2.5$ (98.24%).

## 5 CONCLUSION

In this paper, we introduced a novel representation for AI-generated image detection, termed Semantic-Aware Reconstruction Error (SARE), that quantifies the semantic difference between an image and its caption-guided reconstruction. By effectively leveraging the relationship between an image and its caption, SARE provided a discriminative and generalizable feature for detecting fake images across diverse generative models. Our experimental results demonstrated that SARE significantly improved detection performance in both ID and OOD settings, surpassing existing baselines.

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

# APPENDIX

## A  MORE ANALYSIS OF CAPTIONING MODELS

In Section 4.5, we analyzed the impact of different captioning models on detection performance. To further examine the differences between BLIP (Li et al., 2022) and LLaVA-NeXT (Liu et al., 2024), we present additional analysis based on LPIPS scores (Zhang et al., 2018) and visualizations.

**LPIPS-Based Semantic Shift Analysis**    We measured the LPIPS scores between original images and their caption-guided reconstructions. Figure 7a shows the average LPIPS scores of real and fake images in each subset of the GenImage dataset (Zhu et al., 2023) under two conditions: (1) using captions generated by BLIP, and (2) using captions generated by LLaVA-NeXT. In both cases, real images consistently yield higher LPIPS scores than fake images, suggesting that caption-guided reconstruction serves as a reliable cue for detection. However, as shown in Table 7b, the LPIPS score gap between real and fake images, denoted as $\Delta$ in Eq. 7, is smaller with LLaVA-NeXT than with BLIP, which explains the slightly lower performance reported in Table 5.

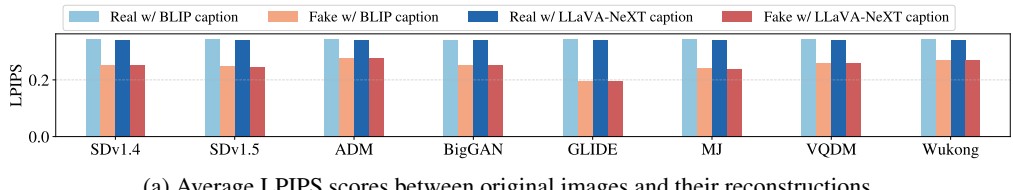

(a) Average LPIPS scores between original images and their reconstructions.

| $\Delta$ | SDv1.4 | SDv1.5 | ADM | BigGAN | GLIDE | MJ | VQDM | Wukong |
|---|---|---|---|---|---|---|---|---|
| w/ BLIP caption | 0.091 | 0.098 | 0.065 | 0.091 | 0.147 | 0.105 | 0.082 | 0.076 |
| w/ LLaVA-NeXT caption | 0.090 | 0.096 | 0.062 | 0.088 | 0.144 | 0.103 | 0.080 | 0.073 |

(b) $\Delta$ values measuring the LPIPS score gap between real and fake images.

Figure 7: Semantic shift analysis based on LPIPS scores (Zhang et al., 2018). Higher scores indicate lower similarity between the original and reconstructed images. Images are reconstructed under two conditions: (1) using captions generated by BLIP (Li et al., 2022), and (2) using captions generated by LLaVA-NeXT (Liu et al., 2024)

| Method | JPEG (QF=90) | JPEG (QF=80) | JPEG (QF=70) | Scale (0.75) | Scale (1.25) |
|---|---|---|---|---|---|
| GramNet (Liu et al., 2020) | 71.22 | 71.02 | 71.22 | 69.89 | 66.31 |
| Conv-B (Liu et al., 2022) | 71.91 | 71.57 | 71.42 | 75.30 | 75.28 |
| UnivFD (Ojha et al., 2023) | 73.24 | 70.42 | 69.75 | 75.96 | 73.99 |
| DIRE (Wang et al., 2023) | 52.79 | 50.65 | 50.34 | 51.94 | 53.77 |
| DE-FAKE (Sha et al., 2023) | 71.00 | 70.88 | 70.44 | 72.33 | 71.88 |
| NPR (Tan et al., 2024) | 69.44 | 70.94 | 70.77 | 71.27 | 71.20 |
| AIDE (Yan et al., 2025) | 57.22 | 58.50 | 60.45 | 83.13 | 83.79 |
| DRCT (Chen et al., 2024) | 80.97 | 78.06 | 76.18 | 79.51 | 74.75 |
| SARE (ours) | 85.64 | 82.72 | 79.14 | 87.60 | 82.74 |

Table 7: Accuracy (ACC, %) performance of robustness evaluation on the GenImage (Zhu et al., 2023) dataset. QF denotes JPEG quality factor. For NPR (Tan et al., 2024) and AIDE (Yan et al., 2025), the publicly released checkpoints are used.

| Method | JPEG (QF=90) | JPEG (QF=80) | JPEG (QF=70) | Scale (0.75) | Scale (1.25) |
|---|---|---|---|---|---|
| GramNet (Liu et al., 2020) | 80.12 | 81.71 | 81.15 | 69.47 | 69.32 |
| Conv-B (Liu et al., 2022) | 90.53 | 90.17 | 90.43 | 94.15 | 96.90 |
| UnivFD (Ojha et al., 2023) | 85.31 | 82.53 | 81.08 | 85.48 | 82.05 |
| DIRE (Wang et al., 2023) | 70.83 | 63.44 | 59.23 | 78.92 | 80.61 |
| DE-FAKE (Sha et al., 2023) | 79.89 | 79.02 | 77.96 | 81.09 | 78.10 |
| NPR (Tan et al., 2024) | 75.23 | 76.54 | 77.09 | 90.87 | 92.12 |
| AIDE (Yan et al., 2025) | 75.23 | 78.12 | 80.66 | 96.28 | 96.00 |
| DRCT (Chen et al., 2024) | 88.89 | 86.18 | 84.05 | 89.29 | 87.93 |
| SARE (ours) | 94.33 | 93.10 | 89.68 | 94.97 | 92.77 |

Table 8: AUC (%) performance of robustness evaluation on the GenImage (Zhu et al., 2023) dataset. QF denotes JPEG quality factor. For NPR (Tan et al., 2024) and AIDE (Yan et al., 2025), the publicly released checkpoints are used.

**Additional Visualizations** Figures 9–12 present real and fake images from the GenImage dataset and their reconstructions guided by captions from BLIP and LLaVA-NeXT. To ensure a fair comparison, real and fake images are selected from the same ImageNet class label. The visualizations show that real images typically undergo larger semantic shifts than fake images during caption-guided reconstruction with both captioning models. In some real image cases, however, the detailed descriptions provided by LLaVA-NeXT yield reconstructions that remain relatively closer to the original input, whereas the concise captions generated by BLIP tend to produce larger shifts.

## B ADDITIONAL ABLATION STUDY

**Influence of *Strength* Parameter** To evaluate the influence of the *strength* parameter on detection performance, we conducted an ablation study on the GenImage dataset by varying the *strength* value from 0.3 to 0.9. Figure 8 shows the accuracy and AUC performance for each *strength* value. The results demonstrate that the model maintains stable performance across all values. In particular, the highest accuracy is obtained at *strength* = 0.5, while the best AUC is achieved at *strength* = 0.7.

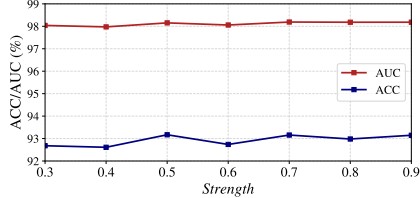

Figure 8: Ablation study on the *strength* parameter conducted on the GenImage dataset (Zhu et al., 2023).

## C LLM USAGE

Large language models (LLMs) were used solely for polishing the writing of this paper.

**Real Images**

Figure 9: Real images from the GenImage dataset (Zhu et al., 2023) with captions from BLIP (Li et al., 2022) and LLaVA-NeXT (Liu et al., 2024), their corresponding reconstructions, and LPIPS scores. For LLaVA-NeXT, we used the prompts "Brief description within 50 words." and "Detailed description within 80 words."

## Real Images

Figure 10: Real images from the GenImage dataset (Zhu et al., 2023) with captions from BLIP (Li et al., 2022) and LLaVA-NeXT (Liu et al., 2024), their corresponding reconstructions, and LPIPS scores. For LLaVA-NeXT, we used the prompts "Brief description within 50 words." and "Detailed description within 80 words."

## Fake Images

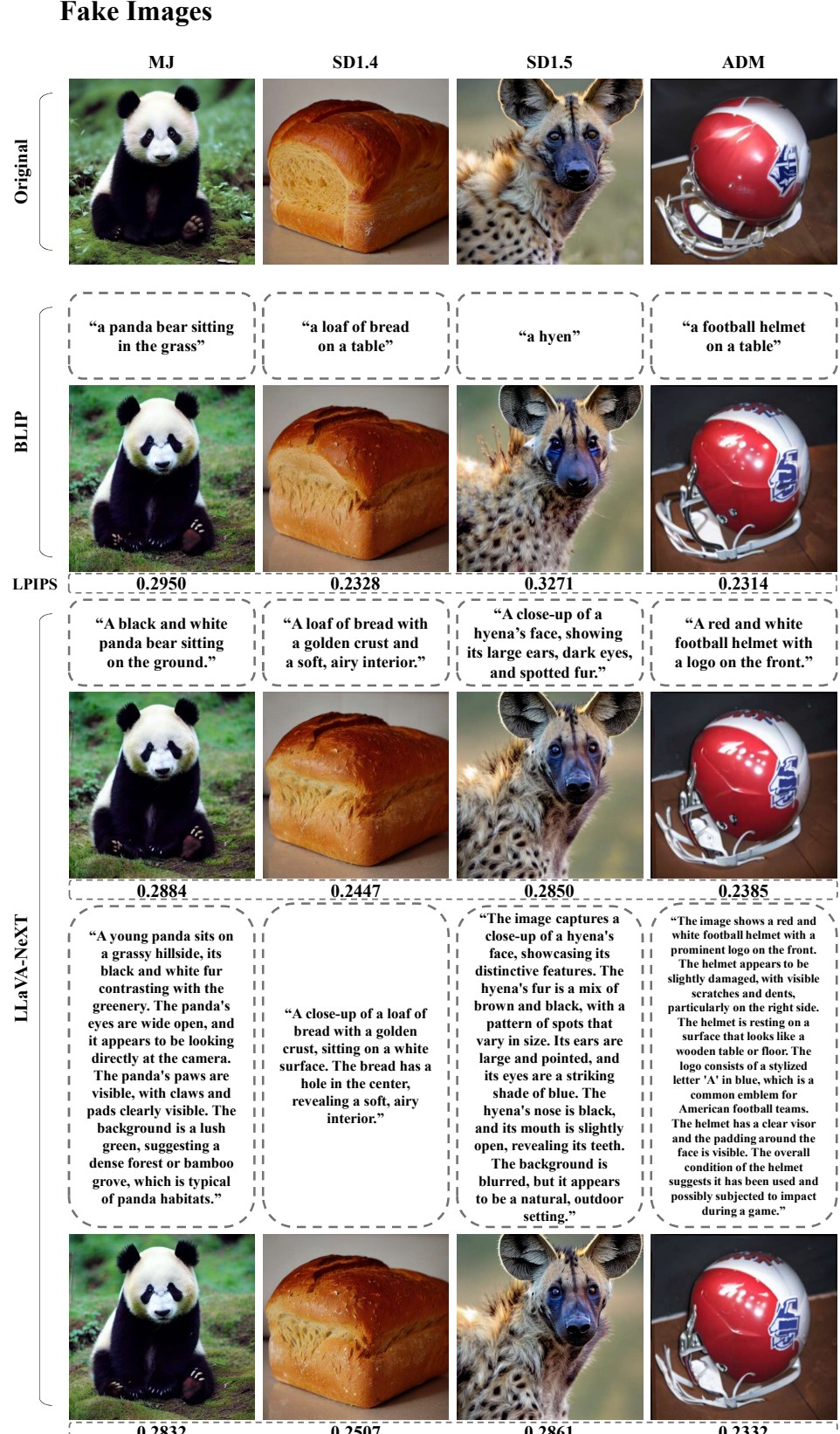

Figure 11: Fake images from the GenImage dataset (Zhu et al., 2023) with captions from BLIP (Li et al., 2022) and LLaVA-NeXT (Liu et al., 2024), their corresponding reconstructions, and LPIPS scores. For LLaVA-NeXT, we used the prompts "Brief description within 50 words." and "Detailed description within 80 words."

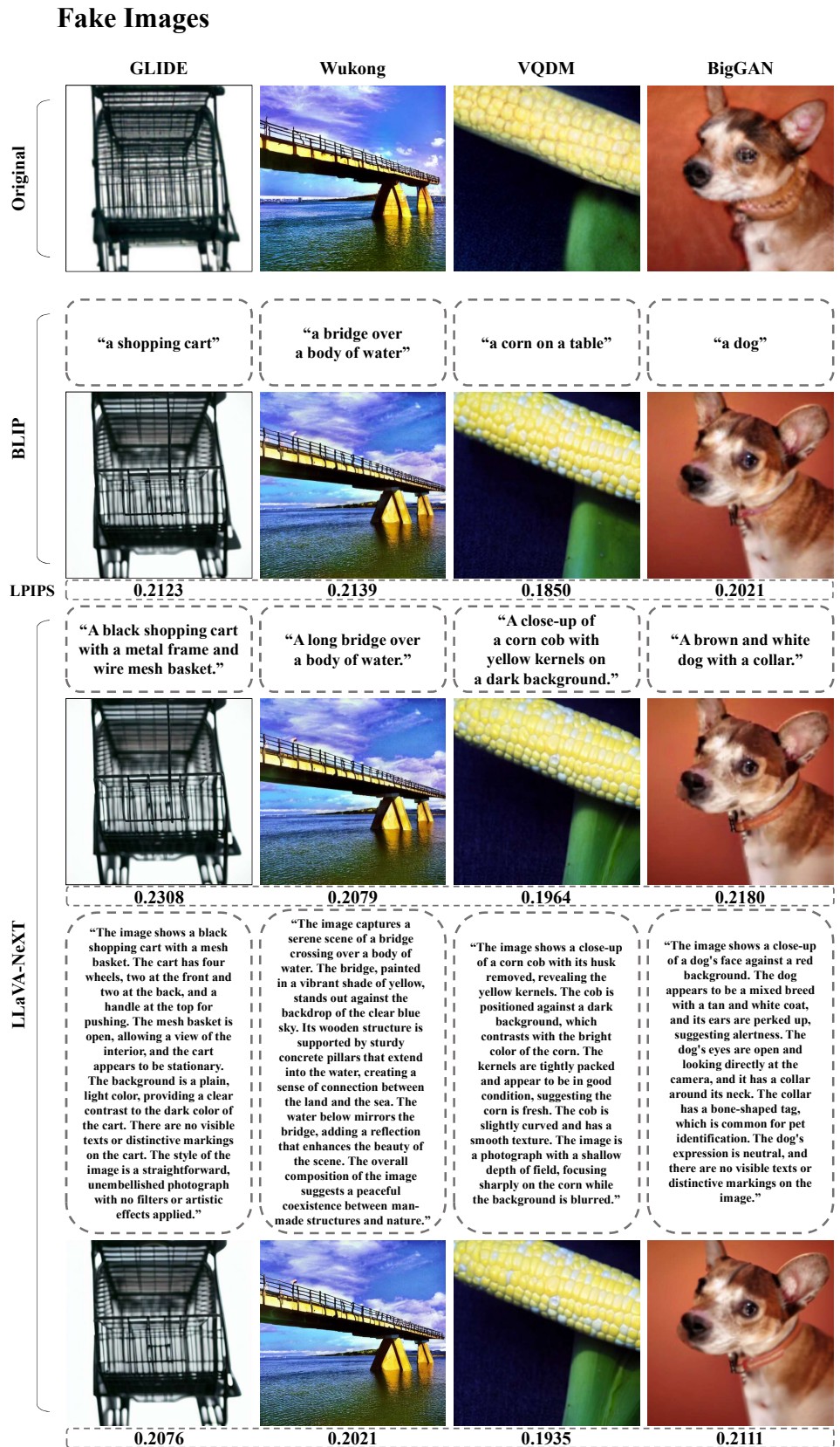

Figure 12: Fake images from the GenImage dataset (Zhu et al., 2023) with captions from BLIP (Li et al., 2022) and LLaVA-NeXT (Liu et al., 2024), their corresponding reconstructions, and LPIPS scores. For LLaVA-NeXT, we used the prompts "Brief description within 50 words." and "Detailed description within 80 words."

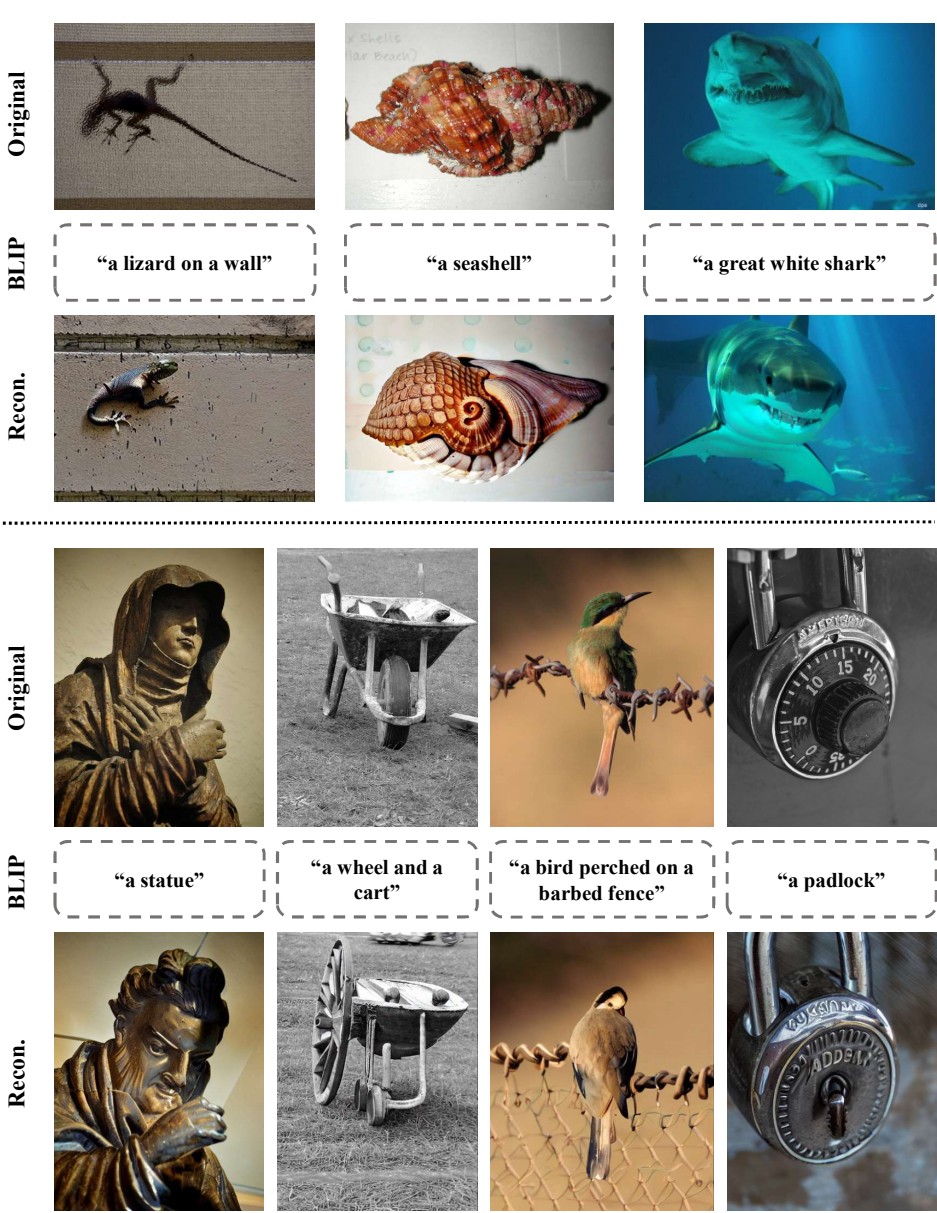

Figure 13: Single object real images from the GenImage dataset (Zhu et al., 2023) with captions from BLIP (Li et al., 2022) and their reconstructions.

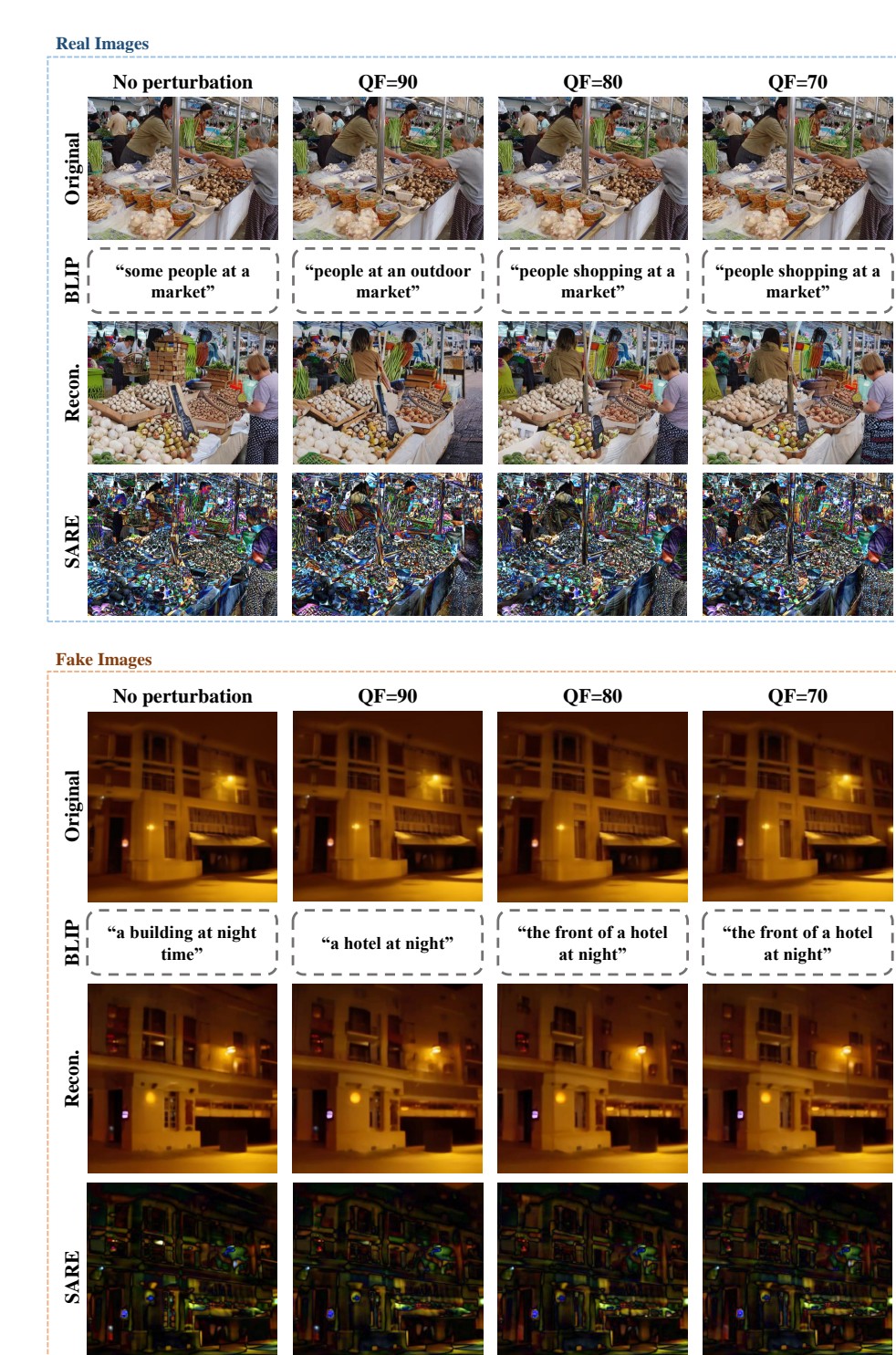

Figure 14: Real and fake images from the GenImage (Zhu et al., 2023) dataset under different levels of JPEG compression (QF = 90, 80, 70), along with their captions generated by BLIP (Li et al., 2022) and caption-guided reconstructions.

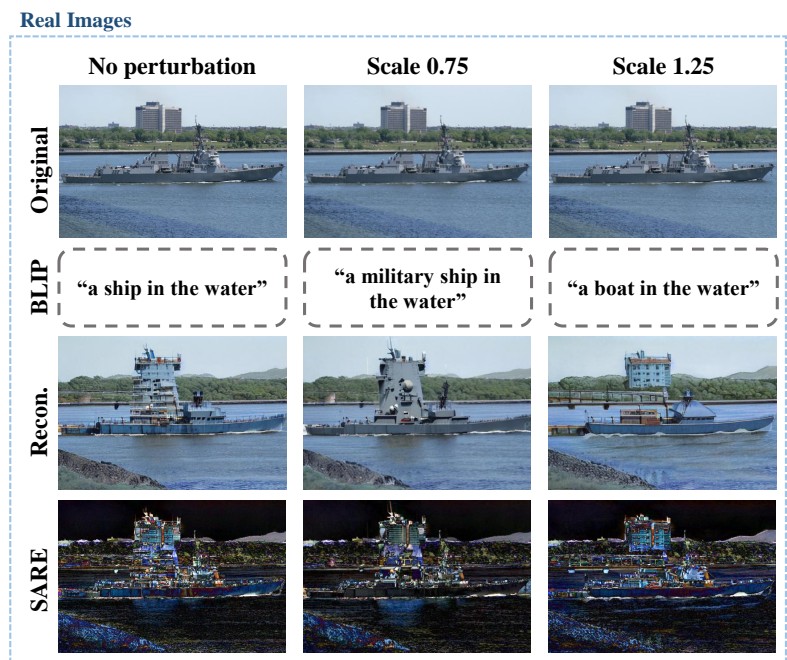

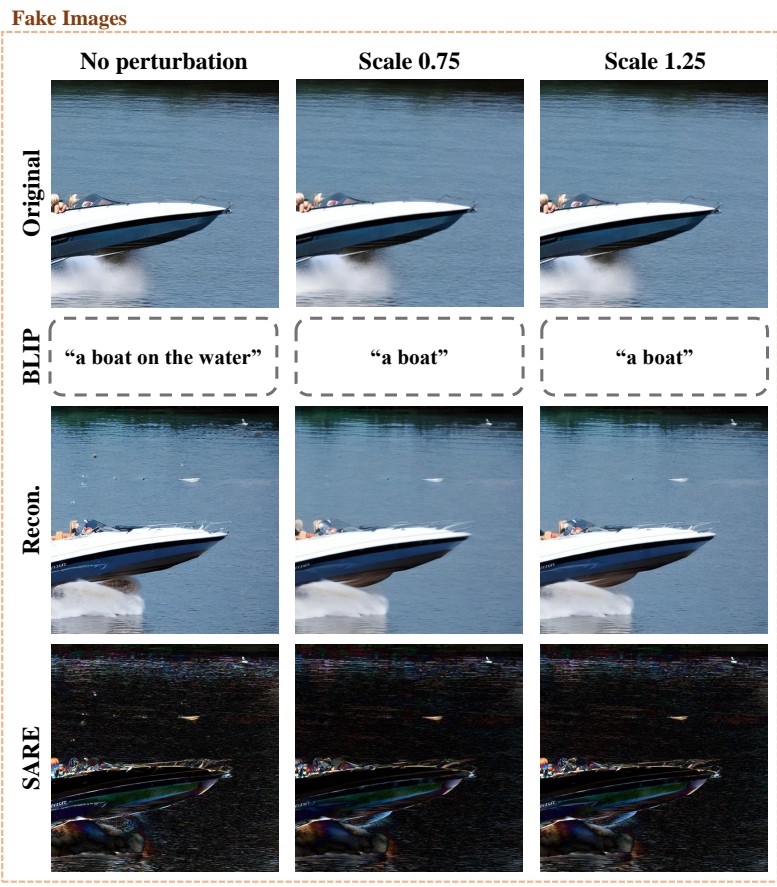

Figure 15: Real and fake images from the GenImage (Zhu et al., 2023) dataset under different levels of scaling (0.75, 1.25), along with their captions generated by BLIP (Li et al., 2022) and caption-guided reconstructions.

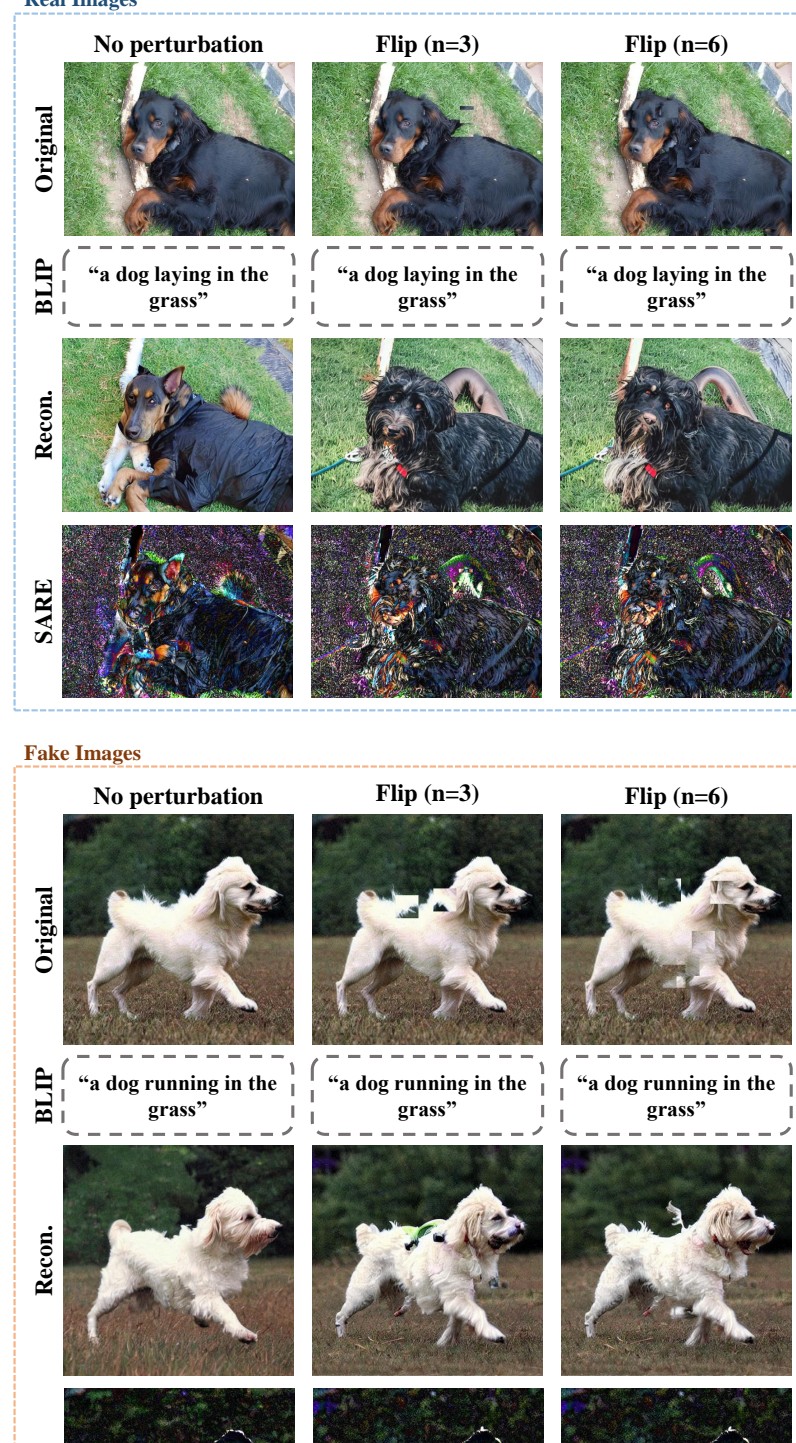

Figure 16: Real and fake images from the GenImage (Zhu et al., 2023) dataset under flip perturbations (3 or 6 regions per image), along with their captions generated by BLIP (Li et al., 2022) and caption-guided reconstructions.

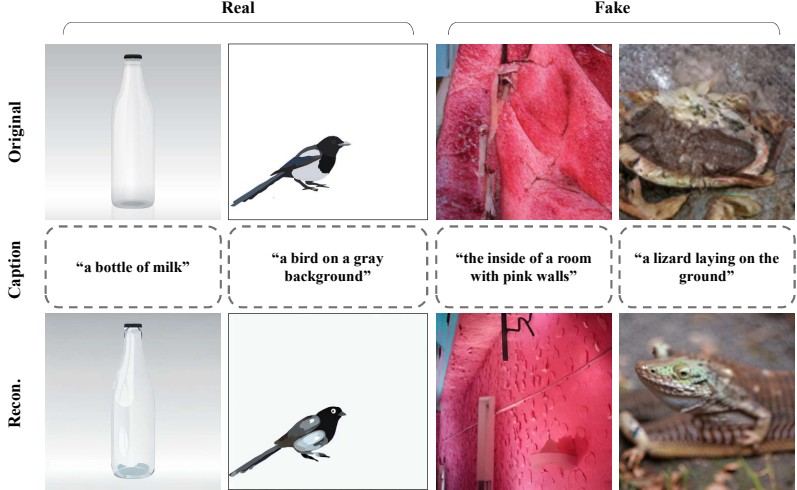

Figure 17: Failure cases of caption-guided reconstructions for real and fake images from the Gen-Image (Zhu et al., 2023) dataset.

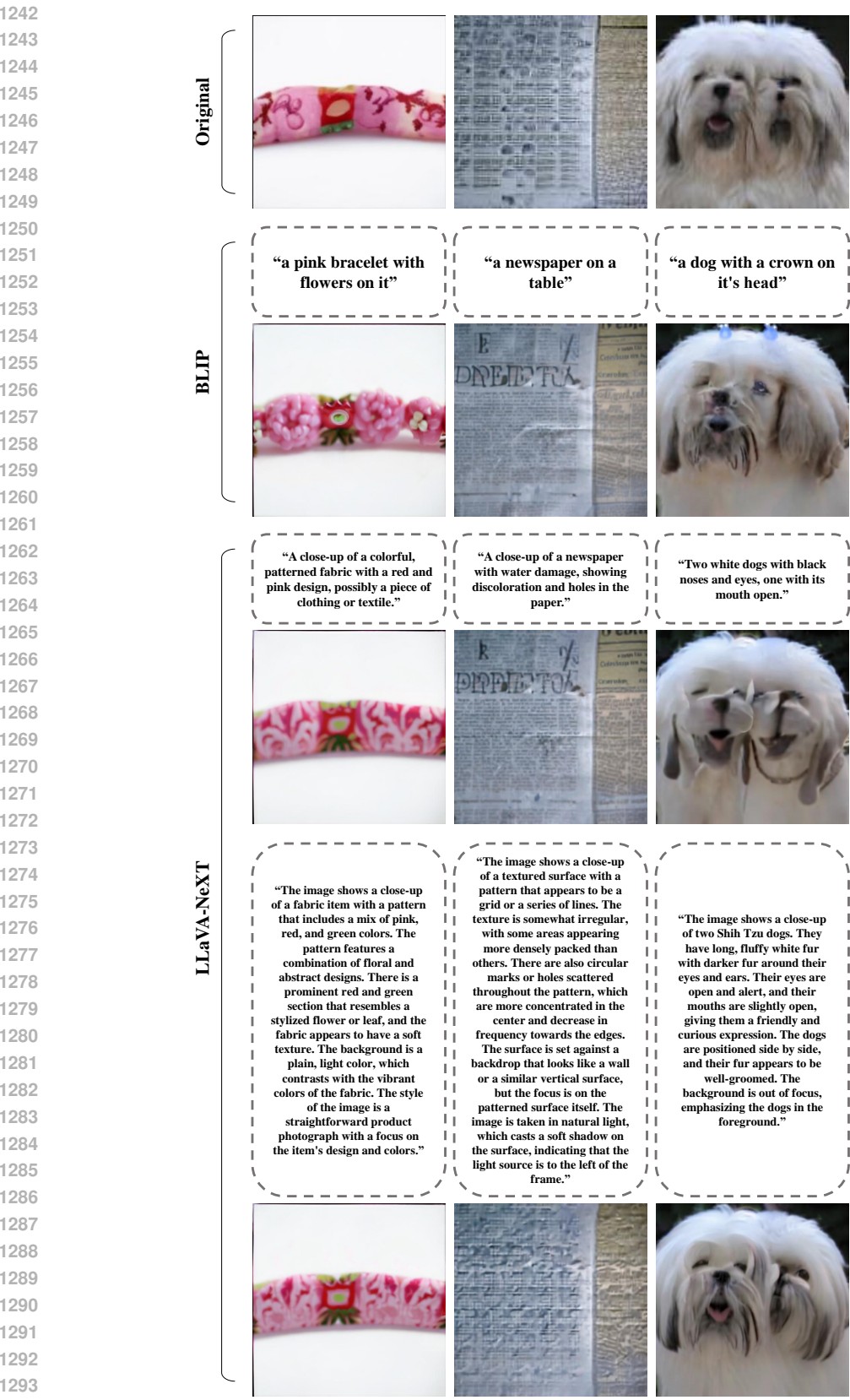

Figure 18: Fake images containing artifacts and distortions from the GenImage (Zhu et al., 2023) dataset and their caption-guided reconstructions.

