# OpenReview forum: "SARE: Semantic-Aware Reconstruction Error for Generalizable AI-Generated Image Detection"
_ICLR.cc/2026/Conference — Submitted to ICLR 2026_

### Official Review · Reviewer_yPPa · 2025-10-16

**Soundness:** 3
**Presentation:** 3
**Contribution:** 2
**Rating:** 4
**Confidence:** 3

**Summary:**

The paper proposes a method Semantic-Aware Reconstruction Error to detect AI-generated images. The method measure the semantic discrepancy between the original image and its caption-guided reconstruction. A pretrained image-captioning model is used to produces the caption of the input image then a stable diffusion is used to reconstruct. The difference between the original image and the reconstructed image is used to classify whether the image is AI-generated.

**Strengths:**

1. The paper is well written and the author provides the source code which increase the soundness of the paper.
2. The proposed method shows strong performance compared to the baseline methods.
3. Although I have concerns about the cost and robustness of the proposed method, the authors have moved beyond the perspective of visual artifacts and demonstrated a new direction.
4.Ablation study is comprehensive

**Weaknesses:**

1. The major concern is computational cost, the proposed method need one time caption generation and one time diffusion reconstruction. The cost is significantly larger than artifact-based or frequency-domain detectors, make it hard to real-time deployment. I also suggest the author add related experiment to show the time cost difference to boost the paper.
2. It seems the method highly rely on the caption model and the reconstruction model. As showing in the experiment,  when using LLaVA-NeXT, the semantic gap between real and fake images becomes smaller. I am not sure the detail setting of caption. What is the length of the caption? A interesting question would be if use stronger caption model and longer sequence of caption, whether this would reduce the semantic difference between the fake and real?  If this conclusion is correct, then the effectiveness of the method seems to rely on the ambiguity of semantics.
3.  Experiments are mainly conducted on GenImage and ForenSynths. I suggest that the authors include a small dataset generated by a VAR model[1], as this model exhibits a larger gap from the reconstruction-based SD. If the proposed method remains effective under this setting, it would greatly strengthen the validity of the proposed method.
4. Once concern is stochasticity in diffusion reconstruction. Since Stable Diffusion sampling is inherently random, the reconstruction result can vary across seeds or sampling parameters. Related ablation is needed.

I will revise my score according to author's reply and other reviewer's opinion.

[1]Tian, Keyu, et al. "Visual autoregressive modeling: Scalable image generation via next-scale prediction." Advances in neural information processing systems 37 (2024): 84839-84865.

**Questions:**

See weakness.

---

> ### Author Response · Authors · 2025-11-20
> **Response to Reviewer yPPa (1/2)**
>
> We sincerely appreciate the reviewer’s thoughtful feedback. We performed additional experiments to address the raised concerns and hope that our responses sufficiently clarify them.
>
> >**Q1. Computation cost analysis**
>
> Thank you for the valuable comment. We provide a computation cost analysis, reporting the FLOPs for each component of the detection pipeline. For diffusion reconstruction, we use Stable Diffusion v1 with 50 total sampling steps. For DIRE, we follow the original configurations of prior unconditional reconstruction-based methods (DRCT, DIRE) and set the strength to 1. For our method, we report FLOPs across different strength values. In the paper, we use a default strength of 0.5.
>
> |GFLOPs| Method              | Backbone | Image Captioning | Diffusion Reconstruction | Total      |
> |-|---------------------|----------|------------------|---------------------------|------------|
> || GramNet            | 2.27     | –                | –                         | 2.27       |
> || Conv-B             | 15.35    | –                | –                         | 15.35      |
> || UnivFD             | 51.9     | –                | –                         | 51.9       |
> || DIRE               | 4.13     | –                | 35675.14                  | 35679.27   |
> || DE-FAKE            | 4.89     | 87.86            | –                         | 92.75      |
> || DRCT               | 51.9     | –                | –                         | 51.9       |
> || SARE (strength=0.3)| 56.03    | 87.86            | 12634.29                  | 12778.18   |
> || SARE (strength=0.4)| 56.03    | 87.86            | 16021.79                  | 16165.68   |
> || SARE (strength=0.5)| 56.03    | 87.86            | 19409.28                  | 19553.17   |
>
> The results show that both DIRE and our method involve substantially higher costs than baselines that do not include a reconstruction stage. However, we find that reducing the strength parameter effectively lowers the computational overhead while maintaining stable detection performance.
>
> We agree that improving efficiency is important for real-time deployment. Developing more efficient diffusion samplers or lightweight reconstruction modules will be an important future work, and we plan to explore these approaches in subsequent research.
>
> > **Q2. Discussion on caption quality and the validity of the core hypothesis**
>
> We appreciate the reviewer’s thoughtful concern. We would like to clarify that our hypothesis is based on the observation that, even with stronger captioning models, fully expressing the rich and fine-grained information contained in real-world photos through natural language remains fundamentally difficult. Consequently, even highly detailed captions inevitably capture only a subset of the underlying visual semantics of real images. This indicates that the semantic discrepancy leveraged by SARE arises from the inherent visual complexity of real images, not from caption incompleteness.
>
> To validate this, we generated long and highly descriptive captions using LLaVA-NeXT with the prompt “Detailed description within 80 words.”. The resulting captions and reconstructions are added in Figure 9-12 in the Appendix, and the corresponding performance is summarized below. SARE achieves performance comparable to the BLIP with these more detailed captions. Notably, the longest caption setting even outperforms the medium-length setting.
>
> |Image Captioning|Prompt|Avg. length (words)|Avg. ACC (%)|Avg. AUC (%)|
> |-|-|-|-|-|
> |BLIP|-|5.75|93.17|98.15|
> |LLaVA-NeXT|"Brief description within 50 words"|13.78|92.51|97.95|
> ||"Detailed description within 80 words"|70.87|92.88|98.01|
>
> We also computed the average LPIPS scores on the GenImage dataset across all captioning settings. While LPIPS values decrease for both real and fake images as captions become longer, the LPIPS gap between real and fake images becomes larger in the longest caption setting than in the medium-length setting, which aligns with the improved detection performance.
> *These findings demonstrate that leveraging detailed captions does not necessarily reduce the semantic shift gap between real and fake images.*
>
> ||Avg. LPIPS|delta|
> |-|-|-|
> |Real w/ BLIP|0.3437||
> |Fake w/ BLIP|0.2494|0.0943|
> |Real w/ medium LLaVA-NeXT|0.3407||
> |Fake w/ medium LLaVA-NeXT|0.2486|0.0921|
> |Real w/ long LLaVA-NeXT|0.3406||
> |Fake w/ long LLaVA-NeXT|0.2483|0.0923|

---

> ### Author Response · Authors · 2025-11-20
> **Response to Reviewer yPPa (2/2)**
>
> >**Q3. Experiments on VAR-generated dataset**
>
> Thank you for the thoughtful suggestion. Following the reviewer’s recommendation, we evaluated our method on a dataset generated by the VAR [1] model. We constructed a test set using 6,000 real images from ImageNet (6 images per class) and 6,000 fake images generated by VAR using the corresponding ImageNet class labels. We performed a cross-dataset evaluation, where the detectors were trained on the SDv1.4 subset of GenImage and tested on the VAR-generated dataset. SARE consistently outperforms the baseline under this setting and achieves the best performance among all detectors, demonstrating its robustness.
>
>
> VAR| Method   | ACC (%)   | AUC (%)   |
> |-|----------|---------|---------|
> || GramNet  | 51.51  | 62.18  |
> || Conv-B    | 53.79  | 86.06  |
> || UnivFD   | 55.53  | 73.93  |
> || DIRE    | 50.08  | 52.72  |
> || DE-FAKE   | 50.77  | 67.20  |
> || DRCT     | 66.85  | 80.50  |
> || SARE | 68.06 | 86.67 |
>
> >**Q4. Experiments on stochasticity in diffusion reconstruction**
>
> To assess the impact of stochasticity, we generated caption-guided reconstructions for the GenImage test set three times with different random seeds and report both the average and the best performance. The results show that SARE consistently outperforms the baseline even when averaging across seeds.
>
> Stochasticity|Method|Avg. ACC (%) | Avg. AUC (%)|Best ACC (%) | Best AUC (%)|
> |-|-|-|-|-|-|
> ||SARE|92.43 | 97.97|93.17 | 98.15|
>
> In addition, we provide ablations on reconstruction parameters that influence stochasticity. Section 4.5 (Table 5) includes an ablation on the guidance scale, and Appendix B presents an ablation on the strength parameter, demonstrating the stability of SARE across different configurations.
>
> [1] Tian, Keyu, et al. "Visual autoregressive modeling: Scalable image generation via next-scale prediction." Advances in neural information processing systems 37 (2024): 84839-84865.

---

> > ### Author Response · Authors · 2025-11-27
> > **Reach out to Reviewer yPPa**
> >
> > Dear Reviewer yPPa,
> >
> > Thank you for your thoughtful and constructive feedback. We sincerely appreciate the time and effort you have invested in reviewing our work. We have provided our responses to your comments, and we would be glad to address any further concerns or questions you may have.
> >
> > As the rebuttal period is approaching its end, we want to kindly check whether any additional clarification from our side might be useful. Our intention is to fully understand your perspective and further improve the quality of our paper based on your insights.
> >
> > Thank you again for your time and consideration. We would greatly appreciate any additional comments you may wish to share.

---

> > > ### Comment · Reviewer_yPPa · 2025-11-27
> > > **Thanks for the response and efforts**
> > >
> > > Thanks to the authors for the clarifications and for taking the time to respond. I have carefully read the responses as well as the comments from the other reviewers. The replies to Q2 and Q3 have addressed my concerns.
> > >
> > > I appreciate the authors’ effort to explore an alternative approach beyond artifact-based methods. However, given the significantly higher computational cost, I remain doubtful about the feasibility of deploying this method at scale in practical applications. Therefore, I am inclined to maintain my current score.

---

> > > > ### Author Response · Authors · 2025-12-01
> > > > **Thank you for the feedback! Response to Reviewer yPPa**
> > > >
> > > > Thank you for raising this concern.
> > > >
> > > > In our computation cost analysis, we found that the diffusion reconstruction stage accounts for the majority of the overall cost. To address this issue, we conducted an additional experiment using a more efficient reconstruction strategy. Specifically, we adopted the SwiftEdit reconstruction model [1], which leverages a one-step inversion framework and a lightweight backbone network.
> > > >
> > > > The results on the GenImage dataset show that SARE continues to provide clear improvements over baseline detectors even under this efficient reconstruction setting. Importantly, the computational cost of the diffusion reconstruction stage is significantly reduced. These findings suggest that lightweight reconstruction techniques can effectively mitigate the computational burden of SARE, making the method more practical for real-world deployment.
> > > >
> > > > |GFLOPs| Method        | Backbone | Image Captioning | Diffusion Reconstruction | Total      |
> > > > |-|---------------------|----------|------------------|---------------------------|------------|
> > > > || GramNet            | 2.27     | –                | –                         | 2.27       |
> > > > || Conv-B             | 15.35    | –                | –                         | 15.35      |
> > > > || UnivFD             | 51.9     | –                | –                         | 51.9       |
> > > > || DIRE               | 4.13     | –                | 35675.14                  | 35679.27   |
> > > > || DE-FAKE            | 4.89     | 87.86            | –                         | 92.75      |
> > > > || DRCT               | 51.9     | –                | –                         | 51.9       |
> > > > || SARE (SDv1) (strength=0.3)| 56.03    | 87.86        | 12634.29                  | 12778.18   |
> > > > || SARE (SDv1) (strength=0.5)| 56.03    | 87.86        | 19409.28                | 19553.17   |
> > > > || **SARE (SwiftEdit)**| **56.03**    | **87.86**        | **26.46**                | **170.35**   |
> > > >
> > > > |Method|Avg. ACC (%) | Avg. AUC (%)|
> > > > |-|-|-|
> > > > |DRCT|88.81|95.66|
> > > > |SARE (SwiftEdit)|91.50|97.65|
> > > >
> > > > [1] Nguyen et al. SwiftEdit: Lightning Fast Text-Guided Image Editing via One-Step Diffusion. CVPR, 2025.

---

### Official Review · Reviewer_xzkZ · 2025-10-24

**Soundness:** 2
**Presentation:** 3
**Contribution:** 2
**Rating:** 4
**Confidence:** 4

**Summary:**

This paper introduces SARE (Semantic-Aware Reconstruction Error), a novel and generalizable method for detecting AI-generated images. The key idea is to measure the semantic difference between an image and its caption-guided reconstruction, leveraging the observation that real images often undergo larger semantic shifts than fake ones during reconstruction due to their richer, under-described details. Experimental results demonstrate that the proposed method achieves strong generalization and outperforms existing baselines.

**Strengths:**

1. This paper introduces SARE, a novel method that quantifies the semantic gap between an image and its reconstruction given a generated caption, rendering SARE a robust and broadly applicable feature for detection tasks.
2. The paper presents the model architecture and training pipeline of SARE with exceptional clarity, ensuring high reproducibility.

**Weaknesses:**

Major Weaknesses:

1. Previous studies such as [1] have proven that JPEG compression significantly affects AI-generated image detection. The two early-stage datasets selected by the authors fall into this category. Therefore, robustness experiments against various perturbations are highly necessary; the absence of such experiments would greatly undermine the validity of the results presented in Tables 1–4.
2. Previous studies [2] and empirical evidence have shown that classifiers trained with cross-entropy almost saturate on seen classes (e.g., Conv-B and DIRE on SD 1.4 in Table 1, and the same behavior is reported in Table 4 of [2]). By contrast, the proposed method does not reach such near-perfect accuracy on SD 1.4, while delivering much larger gains on the remaining classes. This discrepancy requires a convincing explanation from the authors.
3. Figures 11 and 12 show that fake-image reconstructions stay visually close to the originals.
In Figure 4(a), however, the LPIPS score of fake images under caption guidance is much higher than that of real images reconstructed without caption. This seems to contradict the intuition that real photos should be harder to reconstruct even when no guidance. I recommend that the authors include the corresponding SARE visualizations in Figure 2 to clarify this issue.

Minor Weakness:

a. The authors introduce a novel fusion module, yet no ablation is provided to verify its individual contribution (vs DIRE with semantic-aware reconstruction).

b. In the tables, the authors prefix “+ SARE” to the results. Since SARE is a distinct method, for which DRCT is merely used as a backbone, the plus sign should be removed to avoid the impression that this is an incremental extension.

c. Another limitation arises when the image content is overly complex, containing numerous objects or intricate relationships. In such cases, the caption guidance may struggle to fully capture the scene's nuances, leading to suboptimal reconstruction.


[1] Grommelt et al. Fake or JPEG? Revealing Common Biases in Generated Image
Detection Datasets. ECCV 2024.

[2] Yan et al. A Sanity Check for AI-generated Image Detection. ICLR 2025.

**Questions:**

As described in Weaknesses. My two main concerns are: 1) the JPEG compression experiments, and 2) explanation of the unexplained anomalous results.

---

> ### Author Response · Authors · 2025-11-20
> **Response to Reviewer xzkZ (1/2)**
>
> We sincerely thank the reviewer for the insightful feedback. We appreciate the recognition of the strengths of our work and provide responses to the reviewer’s comments and questions below.
>
> >**Q1. Robustness experiments**
>
> We evaluated the robustness of the detectors with two perturbation methods: JPEG compression (quality factors of 70, 80, 90), and resizing (scales of 0.75, 1.25). The perturbations were applied to the test images of the GenImage dataset. The accuracy results show that our method consistently outperforms the baseline and achieves superior performance compared to other detectors. The AUC results are provided in Table 8 in the Appendix.
>
> |ACC (%)| Method      | JPEG (QF=90) | JPEG (QF=80) | JPEG (QF=70) | Scale (0.75) | Scale (1.25) |
> |-|-------------|--------------|--------------|--------------|--------------|--------------|
> || GramNet     | 71.22 | 71.02 | 71.22 | 69.89 | 66.31 |
> || Conv-B      | 71.91 | 71.57 | 71.42 | 75.30 | 75.28 |
> || UnivFD      | 73.24 | 70.42 | 69.75 | 75.96 | 73.99 |
> || DIRE        | 52.79 | 50.65 | 50.34 | 51.94 | 53.77 |
> || DE-FAKE     | 71.00 | 70.88 | 70.44 | 72.33 | 71.88 |
> || NPR         | 69.44 | 70.94 | 70.77 | 71.27 | 71.20 |
> || AIDE        | 57.22 | 58.50 | 60.45 | 83.13 | 83.79 |
> || DRCT        | 80.97 | 78.06 | 76.18 | 79.51 | 74.75 |
> || SARE  | 85.64 | 82.72 | 79.14 | 87.60| 82.74 |
>
>
> We further visualized SARE under the perturbation settings in Figure 14, 15 in the Appendix. The visualizations show that SARE remains stable even when distortions are applied, and real images consistently exhibit larger SARE values than fake images.
>
> >**Q2. Explanation for not reaching near-perfect accuracy on SDv1.4**
>
> In our work, we use a frozen CLIP: ViT-L/14 image encoder, following the settings of prior works, DRCT and UnivFD. Unlike detectors that fully train their image encoder, such as Conv-B and DIRE, methods with a frozen encoder (including UnivFD, DRCT, DE-FAKE, and our method) consistently do not saturate on the seen generator used for training. This phenomenon is also observed in Table 4 of [1], where UnivFD and GenDET, both of which use frozen encoders, do not achieve near-perfect scores.
>
> >**Q3. SARE visualizations**
>
> In response to the reviewer’s suggestion, we have added SARE visualizations in Figure 2. Since DIRE is based on unconditional diffusion reconstruction, it primarily captures artifact-level differences. In contrast, SARE leverages caption-guided reconstruction, which tends to produce larger differences than DIRE. Notably, despite these larger values, SARE consistently exhibits higher reconstruction errors on real images than on fake images across both in-distribution and out-of-distribution generators.
>
> [1] Yan et al. A Sanity Check for AI-generated Image Detection. ICLR 2025.

---

> ### Author Response · Authors · 2025-11-20
> **Response to Reviewer xzkZ (2/2)**
>
> >**Qa. Ablation studies on the individual components of SARE**
>
> To evaluate the effect of caption guidance, we compare DIRE and DIRE with caption guidance on the GenImage dataset. The results show a clear performance improvement, indicating that semantic-aware reconstruction provides a more discriminative cue than unconditional reconstruction.
>
> |Method|Avg. ACC (%) | Avg. AUC (%)|
> |-|-|-|
> |DIRE|70.35|83.01|
> |DIRE w/ semantic-aware recon.|81.67|94.29|
>
> We additionally performed an ablation on the fusion module on the GenImage dataset and observed that the cross-attention design achieves the best performance.
>
> | Fusion module   | Avg. ACC (%) | Avg. AUC (%) |
> |-----------------|---------|---------|
> | Concat          | 91.81   | 97.53   |
> | FiLM [2]            | 89.77   | 98.12   |
> | Cross Attention | 93.17 | 98.15 |
>
> >**Qb. Table modification**
>
> Thank you for the valuable comment. We have revised Tables 1-6 to list SARE as an independent method.
>
> >**Qc. Discussion on caption inaccuracies**
>
> Thank you for the insightful comment. We agree that when an image contains highly complex visual content, caption errors may occur and lead to suboptimal reconstructions for fake images, which could be a potential limitation. However, our core hypothesis is that fake images are generated directly from text prompts and therefore tend to contain only the information explicitly specified in the prompt, leading to inherently high similarity between the image and its caption. In contrast, real images often exhibit rich and complex visual details that cannot be fully captured by a caption, resulting in lower similarity and larger semantic discrepancies.
>
> We further note that stronger captioning models such as LLaVA or GPT-4V can mitigate caption inaccuracies. As shown in Table 5, using LLaVA-NeXT yields substantial improvements over the baseline detector, demonstrating that our method remains effective with stronger captioning models.
>
> [2] Perez, et al. FiLM: Visual Reasoning with a General Conditioning Layer. AAAI, 2018.

---

> > ### Comment · Reviewer_xzkZ · 2025-11-21
> >
> > Thank the authors for detailed responses and the extensive additional experiments provided, which address part of my concerns.
> >
> > In the response to Q1, the JPEG compression results presented in robustness experiments appear almost too good to be true. In my opinion, the significant performance gain of DRCT is attributable to the compression perturbation used during its training. However, I could not find any declaration of this specific augmentation in this paper’s implementation details. I am unclear about the reported performance improvement and would appreciate clarification from the authors.
> >
> > In the response to Q2, the given reference is incorrect. After checking several studies such as DRCT, I accept the author’s rebuttal and believe this is probably an oversight.
> >
> > Furthermore, upon re-examining the manuscript, I have identified several new concerns which could significantly impact my overall evaluation of this work.
> >
> > **1. Clarification on the Novelty and Distinctive Contributions.**
> >
> > The Semantic-Aware Reconstruction method has been previously introduced in FakeInversion [1] (which is also cited in this paper, so I assume that the authors are aware of it). In fact, an ablation study employing the DIRE loss was also conducted in [1] (in the subsection titled ’'Inversions are crucial for generalization’’. Interestingly, a performance degradation was observed in their experiments.)  Consequently, it is my view that the content in Figure 1 and Figures 2 (a) and (b) does not represent novel contributions. The only marginally original element is presented in Figure 2 (c), but its novelty is extremely limited. Please explain how this paper substantively differs from the above setting in prior work [1].
> >
> > **2. Regarding LPIPS Metric and Case Representativeness.**
> >
> > As shown in Figure 4, the LPIPS score for "Fake w/ caption guidance" is also quite high. Therefore, I believe the examples in Figures 11 and 12 may have been selectively chosen. For the fake images, the reconstructions generated with caption guidance still exhibit significant deviations from the original images. It would be helpful to know if there are variance results for LPIPS or if the sample selection was entirely randomized. Additionally, the absence of failure cases in the paper undermines its rigor.
> >
> > Moreover, in Figure 2, SARE is mislabeled as DIRE.
> >
> > [1] George et al. Fakeinversion: Learning to detect images from unseen text-to-image models by inverting stable diffusion. CVPR 2024.

---

> ### Author Response · Authors · 2025-11-23
> **Thank you for the constructive feedback! Response to Reviewer xzkZ (1/2)**
>
> We sincerely thank the reviewer for the thoughtful and detailed feedback, which is highly valuable for improving the clarity and quality of our work.
>
> >**Additional response to Q1**
>
> We sincerely thank the reviewer for the careful examination. We confirm that JPEG compression with random quality and random scaling were included in our training augmentation pipeline, and we have updated the manuscript to clearly describe these details. The models reported in the robustness table (GramNet, Conv-B, UnivFD, DIRE, DE-FAKE, DRCT, and SARE) were trained using the same preprocessing and augmentation settings, including JPEG compression and random scaling, ensuring a fair comparison across methods. For AIDE and NPR, which were added during the rebuttal, we used the official checkpoints provided by the authors.
>
> >**Additional response to Q2**
>
> Thank you for pointing this out. We have corrected the reference to [1] in the rebuttal, and we appreciate the reviewer’s careful attention and consideration.
>
> >**Clarification on the novelty and distinctive contributions**
>
> Thank you for the thoughtful comment. We would like to clarify our method from three key aspects.
>
> >**Role of caption conditioning**
>
> First, caption conditioning plays a different role in our method compared to FakeInversion [2]. FakeInversion empirically shows that text conditioning stabilizes the DDIM inversion process and mitigates catastrophic reconstruction failures (as reported in Appendix D.3 of [2]). In contrast, our work is based on the novel hypothesis that captions inherently induce semantic shifts during reconstruction, and these semantic shifts differ between real and fake images. SARE leverages this semantic discrepancy as the core detection signal, rather than aiming to improve inversion stability or reconstruction quality.
>
> >**Independence from DDIM Inversion**
>
> Second, FakeInversion relies on DDIM inversion to recover the noise map, which serves as a key detection signal. In contrast, SARE only requires a conditional diffusion model capable of incorporating text guidance. To verify this, we additionally evaluated SARE using SDXL [3] instead of SDv1.4 and observed consistent improvements over the baselines, indicating that our approach is not dependent on a specific diffusion architecture.
>
> |Recon. model|Avg. ACC (%)|Avg. AUC (%)|
> |-|-|-|
> |SDv1|93.17|98.15|
> |SDXL|92.22|97.77|
>
> >**Generalizable detection**
>
> Third, FakeInversion utilizes the inversion noise map and the reconstructed image to estimate “Stable Diffusion-likelihood” signal, thereby focusing primarily on detecting diffusion-generated images. In contrast, SARE is based on the semantic relationship between an image and its caption, a property that does not depend on a specific generative model, and can thus generalize to fake images produced by a wide range of generators.
>
> [1] Yan et al. A Sanity Check for AI-generated Image Detection. ICLR, 2025.
>
> [2] George et al. Fakeinversion: Learning to detect images from unseen text-to-image models by inverting stable diffusion. CVPR, 2024.
>
> [3] Podell et al. SDXL: Improving Latent Diffusion Models for High-Resolution Image Synthesis. ICLR, 2024.

---

> ### Author Response · Authors · 2025-11-23
> **Thank you for the constructive feedback! Response to Reviewer xzkZ (2/2)**
>
> >**Regarding LPIPS metric and case representatives**
>
> First, we added the LPIPS scores between each original image and its reconstruction in Figure 2. The results show that the LPIPS scores are considerably higher when caption guidance is applied. Without caption guidance, the reconstruction mainly reflects artifacts inherent to the diffusion process and remains unchanged in terms of semantic content. In contrast, caption-guided reconstruction introduces semantic changes even for fake images, such as the texture of animal fur and background elements (SDv1.4 image), the appearance of buildings (VQDM image), and the appearance of liquid inside a wine glass and the shape of the cup (BigGAN image), leading to substantially higher LPIPS values. This demonstrates that, although the semantic shift for fake images is smaller than that for real images, caption guidance still induces a non-negligible shift compared to the unconditional reconstruction, and the LPIPS scores capture this behavior.
>
> Second, we included the LPIPS scores for the examples in Figure 9-12. Fake images exhibit larger LPIPS values under caption-guided reconstruction than under the unconditional reconstruction results presented in Figure 4. We also provide the variance of LPIPS scores under caption guidance below. To quantify the improvement in separability, we additionally measured Cohen’s d and Fisher’s Discriminant Ratio (FDR). These metrics confirm that caption guidance yields greater separation between the real and fake distributions compared to the unconditional setting.
>
> | LPIPS              | Label | Mean   | Variance |
> |------------------------|-------|--------|----------|
> | w/ caption guidance    | real  | 0.3438 | 0.0076   |
> |                       | fake  | 0.2493 | 0.0069   |
>
> | Method              | Cohen's d | FDR    |
> |------------------------|-----------|--------|
> | w/o caption guidance   | 0.8251    | 0.3404 |
> | w/ caption guidance    | 1.1134    | 0.6198 |
>
> >**Failure cases**
>
> Figure 17 presents several failure cases of caption-guided reconstructions in the GenImage dataset, which may suggest a potential limitation of our approach. A few real images, although labeled as real, lack the detail and information typically found in real-world scenes, and therefore, the caption-guided reconstruction introduces minor semantic changes. In contrast, certain fake images contain highly unrecognizable structures, which result in suboptimal captions and consequently lead to relatively larger semantic shifts during reconstruction.

---

> > ### Comment · Reviewer_xzkZ · 2025-11-25
> >
> > Thank the authors for the thorough and thoughtful responses to my review, which have dispelled my earlier negative impression of the paper.
> >
> > However, I still have some concerns regarding the comparison with FakeInversion:
> >
> > - This work and FakeInversion present an interesting contrast regarding semantic reconstruction. FakeInversion suggests that semantic reconstruction contributes to stability, whereas this paper indicates that it acts more like a perturbation, particularly for real images. So, which conclusion is correct? Could the authors provide some insight for this discrepancy?
> >
> > - The fact that FakeInversion takes into account the original image, noise map, and reconstructed image simultaneously makes me wonder whether it deliberately avoided semantic reconstruction approaches due to prior art. I noticed that the paper mentions conducting ablation studies using a method similar to DIRE—which I believe is a form of semantic reconstruction—and found that it actually degrades performance. Could the authors provide a more in-depth analysis and explanation on this point? I believe this could indeed provide a fundamental understanding of the mechanism and significantly benefit the research community.

---

> > > ### Author Response · Authors · 2025-11-27
> > > **Thank you for the feedback! Response to Reviewer xzkZ**
> > >
> > > Thank you for the insightful comments.
> > >
> > > As stated in Section 3 of FakeInversion, the authors do not apply classifier-free guidance (CFG) during DDIM inversion and sampling, and instead rely on the original conditional update rules. However, prior works [1], [2], [3] have shown that meaningful editing and text-driven modifications in the DDIM process require classifier-free guidance (w>1). Without CFG (i.e., w=1), text conditioning does not produce semantic-level changes. Therefore, in FakeInversion, text conditioning primarily serves to stabilize the inversion and prevent collapse, rather than to induce semantic modifications.
> > >
> > > To further examine this, we applied classifier-free guidance during the DDIM sampling to enable caption-guided reconstruction. We employed the DDIM process of Pix-to-Pix Zero [4] Hugging Face implementation, following the configuration described in Appendix C.2 of FakeInversion (except for the use of CFG). We reconstructed images from the GenImage dataset and measured the average LPIPS scores. The results show that real images tend to exhibit larger semantic shifts than fake images, which is consistent with our observations using Stable Diffusion v1 (the default setting in our paper).
> > >
> > > | LPIPS | Method    		    | Avg. real | Avg. fake | Delta  |
> > > | ------ | -------------------------------- | --------- | ----------- | ------- |
> > > |      | w/o caption guidance	   | 0.0666   | 0.0396   | 0.0270 |
> > > |      | w/ caption guidance (SDv1) | 0.3438   | 0.2493   | 0.0945 |
> > > |      | w/ caption guidance (DDIM) | 0.4857   | 0.3864   | 0.0993 |
> > >
> > > These findings suggest that the text conditioning in FakeInversion cannot be considered a semantic-aware reconstruction used in SARE. We believe that this difference may explain the contrasting observations in these two works.
> > >
> > >
> > > [1] Mokady et al. Null-text Inversion for Editing Real Images using Guided Diffusion Models. CVPR, 2023
> > >
> > > [2] Kang et al. Eta Inversion: Designing an Optimal Eta Function for Diffusion-based Real Image Editing. ECCV, 2024
> > >
> > > [3] Dang et al. VidEvo: Evolving Video Editing through Exhaustive Temporal Modeling. IJCAI, 2025
> > >
> > > [4] Parmar et al. Zero-shot image-to-image translation. ACM TOG, 2023

---

### Official Review · Reviewer_Se9Z · 2025-10-31

**Soundness:** 2
**Presentation:** 2
**Contribution:** 2
**Rating:** 4
**Confidence:** 5

**Summary:**

This paper proposes a representation termed Semantic-Aware Reconstruction Error (SARE), which measures the semantic difference between an image and its caption-guided reconstruction. The key hypothesis is real images are harder to reconstruct by cpation-guided reconstruction process. Experiments demonstrate the effectiveness of the proposed method.

**Strengths:**

1. This paper addresses the AI-generated image detection task from the semantic perspective, which is less explored for now.
2. The experimental results support the effectiveness of the proposed method.
3. This paper is well-organized and easy to follow.

**Weaknesses:**

1. My major concern focuses on the hypothesis of this paper: although I believe the corresponding captions of the generated images are semantically close to the image contents, do the authors consider that there would be artifacts in generated images, such as distortion or blurry, which may have impact on the generated captions? And these artifacts have also been used in many previous work for detection. More directly, did the authors have any theoretical or empirical proof for their hypothesis?
2. The authors state their motivation and make comparisons with DIRE in Section 3.1, which is good. But I still have concern on the novelty of this paper since it seems more like an incremental work based on DIRE, from both motivation and method design perspectives. More justifications are needed to address this.
3. For the experiments part, I suggest the authors add more recent baselines.
4. Can the authors present some visualization of their designed SARE representation? This will be beneficial for direct analysis.
5. From the results, we can see the performance on GAN-generated images is clearly lower than diffusion-generated images. More explanations on this are needed. Does this mean the hypothesis works only for text-to-image diffusion models?

**Questions:**

Please refer to the weakness part. My major concerns are the hypothesis and method design parts. If the authors can address them properly, I will raise my ratings.

---

> ### Author Response · Authors · 2025-11-20
> **Response to Reviewer Se9Z (1/2)**
>
> We sincerely thank the reviewer for the thoughtful and constructive feedback. We appreciate the reviewer’s recognition of the strengths of our work, and we provide our detailed responses to the comments and questions below.
>
> >**Q1. Evaluating the impact of distortions on captions and SARE**
>
> Thank you for the valuable comment. To evaluate the influence of artifacts on our hypothesis, we applied a set of perturbation methods to the GenImage test sets, specifically JPEG compression (quality factors of 70, 80, 90) and resizing (scales of 0.75 and 1.25). For each perturbation, we visualized the resulting captions, caption-guided reconstructions, and SAREs in Figure 14, 15 in the Appendix.
> Although the captions are affected by the perturbations, the semantic changes are minor, and both the reconstructions and the SARE values remain stable across all settings. Importantly, real images consistently exhibit larger SARE values than fake images, even under distortions.
>
> Furthermore, the quantitative results presented below show that our method maintains high performance across all perturbation settings, outperforming the baseline DRCT. These findings suggest that SARE is robust to artifacts and provides a reliable semantic cue under distortions. The AUC results are provided in Table 8 in the Appendix. (For AIDE [2] and NPR [3], which were added during the rebuttal, we used the official checkpoints provided by the authors.)
>
> |ACC (%)| Method      | JPEG (QF=90) | JPEG (QF=80) | JPEG (QF=70) | Scale (0.75) | Scale (1.25) |
> |-|-------------|--------------|--------------|--------------|--------------|--------------|
> || GramNet     | 71.22 | 71.02 | 71.22 | 69.89 | 66.31 |
> || Conv-B      | 71.91 | 71.57 | 71.42 | 75.30 | 75.28 |
> || UnivFD      | 73.24 | 70.42 | 69.75 | 75.96 | 73.99 |
> || DIRE        | 52.79 | 50.65 | 50.34 | 51.94 | 53.77 |
> || DE-FAKE     | 71.00 | 70.88 | 70.44 | 72.33 | 71.88 |
> || NPR         | 69.44 | 70.94 | 70.77 | 71.27 | 71.20 |
> || AIDE        | 57.22 | 58.50 | 60.45 | 83.13 | 83.79 |
> || DRCT        | 80.97 | 78.06 | 76.18 | 79.51 | 74.75 |
> || SARE  | 85.64 | 82.72 | 79.14 | 87.60| 82.74 |
>
> >**Q2. Contributions of our work**
>
> Thank you for raising the concern. We would like to highlight the novelty of our approach from the following aspects.
>
> >**Semantic perspective**
>
> In Section 3.1, we pointed out that reconstruction-based methods such as DIRE primarily rely on model-specific artifacts, which limit their generalization. To address this, our work focuses on the inherent semantic properties of real and fake images and introduces a novel perspective that real images tend to undergo larger semantic changes during the caption-guided reconstruction than synthetic images.
>
> To evaluate the effect of caption guidance, we additionally compared DIRE and DIRE with caption guidance on the GenImage dataset. The results show a clear performance improvement.
>
> |Method|Avg. ACC (%) | Avg. AUC (%)|
> |-|-|-|
> |DIRE|70.35|83.01|
> |DIRE w/ caption guidance|81.67|94.29|
>
> >**Architecture designed for SARE**
>
> We propose a fusion module that effectively incorporates the SARE representation into the detector. This module uses a cross-attention mechanism that allows image features to selectively attend to semantic cues extracted from SARE. We conducted an ablation study on different fusion strategies and showed that the cross-attention-based fusion module achieves the best performance, demonstrating its effectiveness.
>
> | Fusion module   | Avg. ACC (%) | Avg. AUC (%) |
> |-----------------|---------|---------|
> | Concat          | 91.81   | 97.53   |
> | FiLM [1]            | 89.77   | 98.12   |
> | Cross Attention | 93.17 | 98.15 |
>
> >**Strong OOD generalization**
>
> By leveraging semantic-aware reconstruction with an effective detection architecture, our method achieves substantial improvements over existing baselines, particularly on out-of-distribution generators.
>
> [1] Perez, et al. FiLM: Visual Reasoning with a General Conditioning Layer. AAAI, 2018.
>
> [2] Yan, et al. A Sanity Check for AI-generated Image Detection. ICLR, 2025.
>
> [3] Tan, et al. Rethinking the Up-Sampling Operations in CNN-based Generative Network for Generalizable Deepfake Detection. CVPR, 2024.

---

> ### Author Response · Authors · 2025-11-20
> **Response to Reviewer Se9Z (2/2)**
>
> >**Q3. Additional baselines**
>
> Thank you for the suggestion. We evaluated two recent detectors, AIDE [2] and NPR [3], on the GenImage dataset as follows. For both added baselines, we used the official checkpoints provided by the authors.
>
> | Method | Avg. ACC (%) | Avg. AUC (%) |
> |--------|---------|---------|
> | NPR    | 88.98   | 97.28   |
> | AIDE   | 87.14   | 96.78   |
> | SARE | 93.17 | 98.15 |
>
> >**Q4. SARE visualization**
>
> We added visualizations of the proposed SARE in Figure 5.
>
> >**Q5. Performance on GAN-generated images**
>
> Thank you for the thoughtful question. We would like to clarify that our hypothesis is based on the semantic relationship between an image and its caption, which is a property that does not depend on a specific generative model.
> The performance gap between diffusion models and GAN models arises because the detector is trained on SDv1.4 images in our experiments, which leads to stronger performance on diffusion-based generators.
>
> However, from a generalization perspective, our method achieves substantial improvements over existing baselines on out-of-distribution synthetic images, including those generated by GANs. As shown in Tables 1 and 2, the performance on BigGAN is comparable to that on other OOD diffusion models and shows clear gains over all baseline detectors. The results in Tables 3 and 4 further confirm the strong OOD generalization capability of our approach.
>
> [2] Yan, et al. A Sanity Check for AI-generated Image Detection. ICLR, 2025.
>
> [3] Tan, et al. Rethinking the Up-Sampling Operations in CNN-based Generative Network for Generalizable Deepfake Detection. CVPR, 2024.

---

> ### Author Response · Authors · 2025-11-27
> **Reach out to Reviewer Se9Z**
>
> Dear Reviewer Se9Z,
>
> Thank you for your thoughtful and constructive feedback. We sincerely appreciate the time and effort you have invested in reviewing our work. We have provided our responses to your comments, and we would be glad to address any further concerns or questions you may have.
>
> As the rebuttal period is approaching its end, we want to kindly check whether any additional clarification from our side might be useful. Our intention is to fully understand your perspective and further improve the quality of our paper based on your insights.
>
> Thank you again for your time and consideration. We would greatly appreciate any additional comments you may wish to share.

---

> > ### Comment · Reviewer_Se9Z · 2025-11-28
> > **Thanks for the response**
> >
> > Thanks for the response from the authors. The response partially addresses my concerns on Q3 and Q4.
> >
> > * For Q1, I do appreciate the authors' effort on robustness experiments, but I don't think all artifacts can be attributed to "perturbations", such as these artifacts left by the generative model itself, i.e., failure cases, architectures, etc.
> > * For Q2, I understand and appreciate the technical contributions or experimental improvements as the author stated, but it is still hard to agree on its "complete "novelty compared to DIRE, I would say it is more like another perspective to consider the same problem or the same phenomenon. And as stated by Reviewer xzkZ in response, it could be beneficial to provide a fundamental understanding of the mechanism.
> > * For Q5, the authors state their method does not depend on specific model but the results on GAN-generated images are lower than diffusion-generated images. I think more analysis or explanations on why the method drops are needed.
> >
> > I do appreciate the authors' efforts. However, I still have some concerns as stated above.

---

> > > ### Author Response · Authors · 2025-12-01
> > > **Thank you for the feedback! Response to Reviewer Se9Z (2/3)**
> > >
> > > >**Response to Q2**
> > >
> > > >**Comparison to DIRE**
> > >
> > > Thank you for the comment. While both DIRE and SARE adopt reconstruction-based strategies, their objectives, underlying assumptions, and resulting behaviors are fundamentally different.
> > >
> > > DIRE leverages low-level visual artifacts as its primary detection cue. Its key assumption is that diffusion-generated images can be more accurately reconstructed than real images, since both the original and reconstructed samples lie within the same diffusion distribution and therefore share similar low-level visual patterns. Under this assumption, the reconstruction error primarily reflects artifact-level inconsistencies between the input image and its reconstruction. As a result, DIRE is inherently designed for detecting diffusion-generated images and tends to degrade when applied to OOD generators. Importantly, DIRE does not use any form of text conditioning.
> > >
> > > In contrast, SARE leverages semantic-level reconstruction shifts. It is motivated by the observation that fake images typically align more closely with their captions than real images. During caption-guided reconstruction, the caption often fails to fully describe the complex visual content of real images, leading to substantial semantic changes. Fake images, whose content is well aligned with the caption, undergo only minimal semantic shifts. Since this assumption relies on the relationship between an image and its caption, SARE generalizes across diverse OOD generative models. Notably, SARE explicitly incorporates text conditioning through classifier-free guidance.
> > >
> > > >**Role of caption conditioning**
> > >
> > > Compared to FakeInversion [1], caption conditioning plays fundamentally different role in our method. FakeInversion introduces text conditioning to stabilize the DDIM inversion and sampling, preventing catastrophic reconstruction failures (as reported in Appendix D.3 of [1]). In contrast, our method is built on the hypothesis that captions inherently induce semantic shifts during reconstruction, and that the degree of these shifts differs between real and fake images.
> > >
> > > A key technical reason behind this difference is how caption guidance is applied. As stated in Section 3 of FakeInversion, the authors explicitly avoid classifier-free guidance (CFG) and rely solely on the original conditional update rule (i.e., w = 1). Prior works [2–4] consistently show that semantic-level, text-driven modifications emerge only when CFG is used with w > 1 and without CFG, caption conditioning has little semantic influence on the reconstruction trajectory. Thus, in FakeInversion, caption guidance contributes mainly to stabilizing the reconstruction rather than inducing semantic changes, whereas our method applies CFG to allow captions to meaningfully affect the reconstruction process.
> > >
> > > To examine this difference, we introduced CFG into the DDIM sampling process of FakeInversion. We employed the Pix-to-Pix Zero [5] Hugging Face implementation, following the configuration as Appendix C.2 of FakeInversion, except for enabling CFG. Under this setting, the results show that caption-guided sampling produced clear semantic shifts, and real images exhibited larger shifts than fake images, which is consistent with the behavior reported in our paper. This shift pattern is the opposite of the original observation in FakeInversion and directly supports our hypothesis that with CFG, caption guidance induces semantic changes that serve as a signal for detecting generated images.
> > >
> > > | LPIPS | Method    		    | Avg. real | Avg. fake | Delta  |
> > > | ------ | -------------------------------- | --------- | ----------- | ------- |
> > > |      | w/o caption guidance	   | 0.0666   | 0.0396   | 0.0270 |
> > > |      | w/ caption guidance (SDv1) | 0.3438   | 0.2493   | 0.0945 |
> > > |      | w/ caption guidance (DDIM) | 0.4857   | 0.3864   | 0.0993 |
> > >
> > > [1] George et al. Fakeinversion: Learning to detect images from unseen text-to-image models by inverting stable diffusion. CVPR, 2024.
> > >
> > > [2] Mokady et al. Null-text Inversion for Editing Real Images using Guided Diffusion Models. CVPR, 2023
> > >
> > > [3] Kang et al. Eta Inversion: Designing an Optimal Eta Function for Diffusion-based Real Image Editing. ECCV, 2024
> > >
> > > [4] Dang et al. VidEvo: Evolving Video Editing through Exhaustive Temporal Modeling. IJCAI, 2025
> > >
> > > [5] Parmar et al. Zero-shot image-to-image translation. ACM TOG, 2023

---

> > > ### Author Response · Authors · 2025-12-01
> > > **Thank you for the feedback! Response to Reviewer Se9Z (3/3)**
> > >
> > > >**Response to Q5**
> > >
> > > Thank you for pointing this out.
> > >
> > > In the GenImage dataset evaluation (Table 1 and 2) where the models were trained on the Stable Diffusion v1.4 (SDv1.4) subset, SARE achieves strong performance on the BigGAN subset (92.05% ACC, 97.51% AUC), comparable to other OOD diffusion model subsets.
> > >
> > > However, in the cross-dataset evaluation (Tables 3 and 4), where the models were trained on the GenImage SDv1.4 subset and tested on the ForenSynths dataset, we observe a performance drop on the GAN subsets of ForenSynths compared to the BigGAN results in the GenImage evaluation, although our method still shows overall improvements over existing baselines.
> > >
> > > To analyze this, we measured the average real accuracy and fake accuracy for the GAN subsets in ForenSynths. The results show that fake accuracy remains consistently high across the GAN subsets, whereas real accuracy is clearly lower, which primarily accounts for the performance drop observed on GAN subsets. We would like to note that according to Appendix B of [6], the real images in the GAN subsets of ForenSynths were resized, whereas the real images in GenImage were not. In our robustness experiments on GenImage, we confirmed that such resizing (scaling) can affect the model’s performance, which may explain the reduced real accuracy observed in ForenSynths. Notably, even for the BigGAN subset in ForenSynths, which shares the same ImageNet real images and generator with the GenImage BigGAN subset, we observe a performance decrease, further indicating that resizing may influence the results.
> > >
> > > |Method|Avg. Fake ACC (%) | Avg. Real ACC (%)|
> > > |-|-|-|
> > > |SARE|91.96|74.25|
> > >
> > > [6] Wang et al. CNN-generated images are surprisingly easy to spot... for now. CVPR, 2025

---

> ### Author Response · Authors · 2025-12-01
> **Thank you for the feedback! Response to Reviewer Se9Z (1/3)**
>
> We appreciate the reviewer’s insightful comments, which greatly helped us improve the paper.
>
> >**Response to Q1**
>
> Thank you for the insightful comment.
>
> As shown in Appendix Figure 18, we analyzed cases where fake images contain artifacts arising from the intrinsic characteristics of generative models. In such failure cases where the generated images exhibit severe distortions or highly unrecognizable structures, BLIP may produce suboptimal captions, resulting in a semantic shift during reconstruction. In contrast, when we employ a stronger captioning model such as LLaVA-NeXT with prompts like “Detailed description within 80 words” (the detailed settings are provided in the table below), the captions more accurately capture the underlying visual content, leading to a smaller semantic shift.
>
> |Image Captioning|Prompt|Avg. length (words)|Avg. ACC (%)|Avg. AUC (%)|
> |-|-|-|-|-|
> |BLIP|-|5.75|93.17|98.15|
> |LLaVA-NeXT|"Brief description within 50 words"|13.78|92.51|97.95|
> ||"Detailed description within 80 words"|70.87|92.88|98.01|
>
> To further examine the effectiveness of using more detailed captions, we evaluated the detection performance on the GenImage dataset using captions produced by each captioning model. Across all settings, SARE consistently produced clear improvements over the baseline. Additionally, LPIPS measurements showed that using more detailed captions tends to reduce the LPIPS scores by better describing the distortions, while real images still exhibit substantially larger scores than fake images in all cases, confirming the consistency of the semantic shift pattern.
>
> ||Avg. LPIPS|delta|
> |-|-|-|
> |Real w/ BLIP|0.3437||
> |Fake w/ BLIP|0.2494|0.0943|
> |Real w/ medium LLaVA-NeXT|0.3407||
> |Fake w/ medium LLaVA-NeXT|0.2486|0.0921|
> |Real w/ long LLaVA-NeXT|0.3406||
> |Fake w/ long LLaVA-NeXT|0.2483|0.0923|
>
> These findings indicate that artifacts introduced by generative models can be effectively mitigated by leveraging stronger captioning models, and that our core assumption, the larger semantic shift of real images compared to fake ones, remains consistently valid even under these enhanced captioning conditions.
>
> We additionally conducted an experiment using flip perturbations to further assess the robustness of SARE. Specifically, we randomly selected rectangular regions whose width and height correspond to 10% of the original image. We then applied vertical or horizontal flipping to either three or six regions per image. Across all configurations, our method consistently outperformed the baseline, demonstrating robustness to flip perturbations. Moreover, as shown in Appendix Figure 16, fake images consistently exhibit smaller SARE values than real images, even under distortions.
>
> |ACC (%) / AUC (%)| Method      | FLIP (n=3) | FLIP (n=6) |
> |-|-------------|--------------|--------------|
> || DRCT        | 87.10 / 94.29 | 86.63 / 93.93 |
> || SARE  | 91.08 / 96.81 | 90.27 / 96.51 |

---

### Official Review · Reviewer_CduT · 2025-11-01

**Soundness:** 3
**Presentation:** 4
**Contribution:** 3
**Rating:** 4
**Confidence:** 4

**Summary:**

This paper proposes SARE (Semantic-Aware Reconstruction Error), a novel representation for detecting AI-generated images that aims to improve generalization to unseen generative models. The key idea is that real images typically contain more complex semantics than their captions can fully describe, leading to larger semantic shifts when reconstructed through caption-guided diffusion. In contrast, AI-generated images tend to align more closely with their captions, thus showing smaller reconstruction discrepancies. SARE measures this semantic difference and integrates it into the detection backbone through a cross-attention fusion mechanism. Experiments on GenImage and ForenSynths show consistent accuracy and AUC improvements over baselines such as DIRE, DRCT, and DE-FAKE, demonstrating good cross-model generalization.

**Strengths:**

- **Conceptual novelty and fresh perspective.** The work moves beyond artifact-based detection toward a semantic-consistency-based approach, which is an emerging and meaningful direction for robust AI-generated image detection.
- **Empirical performance and generalization.** SARE exhibits strong cross-model robustness, achieving the best average AUC and accuracy across diverse unseen generators (ADM, GLIDE, VQDM, BigGAN).
- **Comprehensive experiments.** The paper provides ablation studies on captioning models (BLIP vs. LLaVA-NeXT), parameter sensitivity (guidance scale, strength), and cross-dataset transfer, ensuring good reproducibility and coverage.

**Weaknesses:**

- Limited validity of the core hypothesis and its dependence on caption quality.
- Dependence on specific reconstruction and detection architectures.
- High computational overhead and lack of efficiency analysis.

**Questions:**

This paper presents an interesting and conceptually novel approach for AI-generated image detection by leveraging semantic-aware reconstruction errors (SARE). The idea of distinguishing real and synthetic images through caption-conditioned semantic shifts is innovative and well motivated from an interpretability standpoint. The experiments are comprehensive, and the visualization results help illustrate the mechanism clearly. However, I have several concerns that should be addressed to strengthen the paper:

**Limited validity of the core hypothesis and its dependence on caption quality.**

The core assumption of SARE—that *real images* exhibit larger semantic shifts because their captions fail to capture full visual details, while *fake images* align closely with their captions—is insufficiently validated and may result from caption incompleteness rather than an inherent semantic property. Most qualitative examples compare complex, multi-object real images with simple synthetic ones, leaving it unclear whether the same trend holds for *simple real images* (e.g., single-object photos).

Moreover, all “real” samples are from GenImage and ForenSynths, a narrow and curated dataset, so the method’s validity for diverse real-world imagery (e.g., faces, artistic photos, noisy web content) remains untested.

Ablation studies further show that SARE performs best with BLIP captions, which are typically shorter and coarser than those from LLaVA-NeXT. Such simplified captions may *artificially exaggerate* semantic gaps for real images, reinforcing the method’s assumption rather than validating it. More critically, this hypothesis could break down when using stronger vision-language models (VLLMs) capable of producing rich, detailed captions. If models like LLaVA, GPT-4V, or Gemini generate semantically complete descriptions that fully cover real image content, the reconstruction-guided semantic gap might vanish—contradicting the core premise of SARE. This suggests that the reported performance gains may primarily arise from *caption simplicity* or *incompleteness bias*, not from genuine semantic differences between real and synthetic images.

**Dependence on specific reconstruction and detection architectures.**

Beyond caption dependence, SARE is closely tied to its reconstruction and detection pipeline.
All reconstructions are performed using Stable Diffusion v1.4, and it remains unclear whether SARE maintains similar effectiveness with other diffusion backbones (e.g., SD v2, ADM, or Flux).
The design also couples a cross-attention fusion mechanism specifically tuned to this pipeline, making it uncertain how well the approach generalizes to other reconstruction paradigms or encoder structures.
Without cross-model validation, the claimed “semantic generalization” may partially reflect architectural bias rather than a universally applicable representation.

**High computational overhead and lack of efficiency analysis.**

The SARE pipeline is computationally heavy, consisting of:
 (i) caption generation via BLIP,
 (ii) text-guided diffusion reconstruction requiring 50 denoising steps, and
 (iii) a cross-attention fusion module for classification.
The paper does not report FLOPs, inference time, or throughput, leaving its scalability uncertain.
For practical deployment—such as social media monitoring or large-scale image verification—this processing cost may severely limit feasibility, despite accuracy gains.

---

> ### Author Response · Authors · 2025-11-20
> **Response to Reviewer CduT (1/2)**
>
> We sincerely appreciate your detailed review with the acknowledgement of our strengths and constructive comments. We conducted additional experiments regarding your questions, and we hope our response can address your concerns.
>
> >**Q1. Limited validity of the core hypothesis and its dependence on caption quality.**
>
> >**1-1. Discussion on the validity of the core hypothesis**
>
> We appreciate the reviewer’s thoughtful concern. We would like to clarify that our hypothesis is based on the observation that, even with stronger captioning models, fully expressing the rich and fine-grained information contained in real-world photos through natural language remains fundamentally difficult. Consequently, even highly detailed captions inevitably capture only a subset of the underlying visual semantics of real images. This indicates that the semantic discrepancy leveraged by SARE arises from the inherent visual complexity of real images, not from caption incompleteness.
>
> To validate this, we generated long and highly descriptive captions using LLaVA-NeXT with the prompt “Detailed description within 80 words.”. The resulting captions and reconstructions are added in Figure 9-12 in the Appendix, and the corresponding performance is summarized below. SARE achieves performance comparable to the BLIP with these more detailed captions. Notably, the longest caption setting even outperforms the medium-length setting.
>
> |Image Captioning|Prompt|Avg. length (words)|Avg. ACC|Avg. AUC|
> |-|-|-|-|-|
> |BLIP|-|5.75|93.17|98.15|
> |LLaVA-NeXT|"Brief description within 50 words"|13.78|92.51|97.95|
> ||"Detailed description within 80 words"|70.87|92.88|98.01|
>
> We also computed the average LPIPS scores on the GenImage dataset across all captioning settings. While LPIPS values decrease for both real and fake images as captions become longer, the LPIPS gap between real and fake images becomes larger in the longest caption setting than in the medium-length setting, which aligns with the improved detection performance.
> *These findings demonstrate that leveraging detailed captions does not necessarily reduce the semantic shift gap between real and fake images.*
>
> ||Avg. LPIPS|delta|
> |-|-|-|
> |Real w/ BLIP|0.3437||
> |Fake w/ BLIP|0.2494|0.0943|
> |Real w/ medium LLaVA-NeXT|0.3407||
> |Fake w/ medium LLaVA-NeXT|0.2486|0.0921|
> |Real w/ long LLaVA-NeXT|0.3406||
> |Fake w/ long LLaVA-NeXT|0.2483|0.0923|
>
> >**1-2. Visualizations of simple real images**
>
> We added qualitative results in Figure 13 in the Appendix using simple, single-object real images from GenImage and their caption-guided reconstructions. Even when the real image contains only a single object, it still includes diverse fine-grained visual details such as texture, pose, or background, which are inherent semantic properties of real images. As a result, caption-guided reconstruction produces noticeable semantic shifts.
>
> >**1-3. Cross-dataset evaluations on diverse real-world images**
>
> To cover more diverse real-world images, we adopted two additional datasets, UniversalFakeDetect [1] and CommunityForensics [2]. UniversalFakeDetect uses LSUN and LAION as real images, where LAION is a non-curated dataset crawled from web pages. CommunityForensics utilizes COCO and FFHQ as real images, where FFHQ consists of high quality face images. We conducted cross-dataset evaluation by training the detectors on the SDv1.4 subset of GenImage and testing on these two datasets. The results show that our method consistently outperforms the baseline across both datasets.
>
> |UniversalFakeDetect|Method|Avg. ACC (%)|Avg. AUC (%)|
> |-|-|-|-|
> ||DRCT|84.43|93.10|
> ||SARE|87.81|95.25|
>
> |CommunityForensics|Method|Avg. ACC (%)|Avg. AUC (%)|
> |-|-|-|-|
> ||DRCT|86.69|94.51|
> ||SARE|88.11|95.37|
>
> [1] Ojha, et al. Towards universal fake image detectors that generalize across generative models. CVPR, 2023.
>
> [2] Park, et al. Community Forensics: Using Thousands of Generators to Train Fake Image Detectors. CVPR, 2025.

---

> ### Author Response · Authors · 2025-11-20
> **Response to Reviewer CduT (2/2)**
>
> >**Q2. Dependence on specific reconstruction and detection architectures.**
>
> >**Ablation studies on reconstruction models and architectures**
>
> We conducted comprehensive ablation studies on reconstruction models and detection architectures using the GenImage dataset.
>
> (1) Reconstruction model: We compared two reconstruction backbones, SDv1 and SDXL, and observed that SARE consistently outperforms the baseline with both models.
>
> |Recon. model|Avg. ACC (%)|Avg. AUC (%)|
> |-|-|-|
> |SDv1|93.17|98.15|
> |SDXL [3]|92.22|97.77|
>
> (2) Backbone detector, semantic encoder, fusion module: We further replaced each component of the detection architecture. Across all configurations, SARE exhibited stable performance and consistently outperformed the baseline detectors.
>
> |Backbone detector|Avg. ACC (%)|Avg. AUC (%)|
> |-|-|-|
> |UnivFD|79.11|90.60|
> |+ SARE|81.35|95.54|
> |DRCT|88.81|95.66|
> |+ SARE|93.17|98.15|
>
> | Semantic Encoder   | Avg. ACC (%)| Avg. AUC (%)|
> |--------------------|---------|---------|
> | EfficientNet-B3 | 93.40 | 98.35 |
> | ConvNeXt-Base      | 92.72   | 98.17   |
> | ResNet50           | 93.17   | 98.15   |
>
> | Fusion module   | Avg. ACC (%) | Avg. AUC (%) |
> |-----------------|---------|---------|
> | Concat          | 91.81   | 97.53   |
> | FiLM [4]            | 89.77   | 98.12   |
> | Cross Attention | 93.17 | 98.15 |
>
> >**Q3. High computational overhead and lack of efficiency analysis.**
>
> >**Computation cost analysis**
>
> Thank you for the valuable comment. We provide a computation cost analysis by reporting the FLOPs for each component of the detection pipeline. For diffusion reconstruction, we use Stable Diffusion v1 with 50 total sampling steps. For DIRE, we follow the original configurations of prior unconditional reconstruction-based methods (DRCT, DIRE) and set the strength to 1. For our method, we report FLOPs across different strength values. In the paper, we use a default strength of 0.5.
>
> |GFLOPs| Method              | Backbone | Image Captioning | Diffusion Reconstruction | Total      |
> |-|---------------------|----------|------------------|---------------------------|------------|
> || GramNet            | 2.27     | –                | –                         | 2.27       |
> || Conv-B             | 15.35    | –                | –                         | 15.35      |
> || UnivFD             | 51.9     | –                | –                         | 51.9       |
> || DIRE               | 4.13     | –                | 35675.14                  | 35679.27   |
> || DE-FAKE            | 4.89     | 87.86            | –                         | 92.75      |
> || DRCT               | 51.9     | –                | –                         | 51.9       |
> || SARE (strength=0.3)| 56.03    | 87.86            | 12634.29                  | 12778.18   |
> || SARE (strength=0.4)| 56.03    | 87.86            | 16021.79                  | 16165.68   |
> || SARE (strength=0.5)| 56.03    | 87.86            | 19409.28                  | 19553.17   |
>
> The results show that both DIRE and our method involve substantially higher costs than baselines that do not include a reconstruction stage. However, we find that reducing the strength parameter effectively lowers the computational overhead while maintaining stable detection performance.
> We agree that improving efficiency is important for real-time deployment. Developing more efficient diffusion samplers or lightweight reconstruction modules will be an important future work, and we plan to explore these approaches in subsequent research.
>
> [3] Podell, et al. SDXL: Improving Latent Diffusion Models for High-Resolution Image Synthesis. ICLR, 2024.
>
> [4] Perez, et al. FiLM: Visual Reasoning with a General Conditioning Layer. AAAI, 2018.

---

> > ### Comment · Reviewer_CduT · 2025-11-24
> >
> > Hi authors, thanks for your detailed rebuttal which has basically addressed most of my concerns. I will raise my score.

---

> > > ### Author Response · Authors · 2025-11-25
> > > **Thank you for your feedback!**
> > >
> > > Thank you for your constructive feedback and for taking the time to assess our work. We sincerely appreciate your efforts and are glad that your feedback has helped further improve the quality of our paper.

---

### Comment · Area_Chair_QhR6 · 2025-11-25

Dear Reviewers,

The authors have submitted their responses to your questions and feedbacks. Please read them and give your comments.

Regards, AC

---

### Meta-Review · Area_Chair_K5qY · 2025-12-30

**Summary:**

This paper presents a novel approach for AI-generated image detection by introducing the Semantic-Aware Reconstruction Error (SARE) representation. The core contribution lies in the hypothesis that real images undergo noticeable semantic shifts during caption-guided reconstruction, while AI-generated images show minimal changes due to their closer alignment with captions.

The reviewers raised several significant concerns that merit careful consideration:
1. Hypothesis Validity and Mechanism: While the authors provide empirical evaluation supporting their hypothesis, the underlying mechanism remains largely unexplored. The fundamental question of why real and fake images exhibit different semantic behaviors during reconstruction is not adequately addressed, leaving the theoretical foundation somewhat incomplete.
2. Innovation and Differentiation: The proposed method's distinction from existing reconstruction-based approaches like DIRE and FakeInversion primarily hinges on the use of caption guidance. However, without a clear mechanistic explanation, the reviewers found it difficult to assess whether this represents a substantial innovation or merely an incremental improvement. The novelty appears limited compared to established methods.
3. Computational Overhead: A major practical concern is the significant computational cost associated with the caption-guided reconstruction process. This overhead substantially impacts the method's practical applicability and deployment feasibility, especially when compared to more efficient alternatives.

The authors have responded to some concerns with additional empirical validation but have not sufficiently addressed the fundamental questions about the mechanism behind their hypothesis. The lack of mechanistic explanation makes it challenging to properly evaluate the true significance of caption guidance compared to other reconstruction methods.

Given these substantial concerns regarding theoretical foundation, innovation level, and practical applicability, the decision is to reject the paper in its current form. The authors are encouraged to continue their investigation into the underlying mechanisms of their hypothesis, which could potentially lead to a more substantial contribution in future work.

**Reviewer Concerns:**

please refer to the above summary

**Reviewer Scores:**

please refer to the above summary

---

### Decision · Program_Chairs · 2026-01-26

Reject